



# Groundwater mean residence time of a sub-tropical barrier sand island

Harald Hofmann[1,5], Dean Newborn[1], Ian Cartwright[2], Dioni I. Cendón[3], and Matthias Raiber[4]

[1]School of Earth and Environmental Sciences, The University of Queensland, St Lucia QLD, Australia
[2]School of Earth, Atmosphere and Environment - Monash University, Clayton VIC, Australia
[3]Australian Nuclear Science and Technology Organization, Lucas Heights NSW, Australia
[4]Commonwealth Science and Industrial Research Organisation (CSIRO), Dutton Park QLD, Australia
[5]Geotechnical Engineering Centre - Civil Engineering - The University of Queensland, St Lucia QLD, Australia

**Correspondence:** Harald Hofmann (h.hofmann@uq.edu.au)

**Abstract.** Fresh groundwater on barrier islands is affected by changing sea levels, groundwater use and precipitation variability due to climate change. These systems are also vulnerable to contamination and groundwater over-abstraction. Constraining groundwater mean residence times (MRT) and flow paths are essential for understanding and managing these resources.

This study uses tritium ($^3$H) and carbon-14 ($^{14}$C) to determine the MRT of groundwater along a bore transect across North

Stradbroke Island, South-East Queensland, Australia. Hydraulic properties, major ion geochemistry and stable isotopes are used to validate residence times, and to identify processes responsible for their variability. $^3$H activities range from <0.01 to 1 TU, which are lower than those of local average rainfall (1.6-2 TU). $^{14}$C concentrations range from 62.5 to 111 pMC. Estimated $^3$H MRT determined using lumped parameter models range from 37 years to >150 years. Recharge occurs over the entire island and groundwater MRT increase vertically and laterally towards the coastal discharge areas. MRT estimated from $^{14}$C display

similar spatial relationships but have a much greater range (modern to up approximately 5,000 years). Water diversion and retention by perched aquifers with underlying lower permeability units in the unsaturated part of the dune systems are so far the most likely course for relatively long MRT. The results indicate that these perched aquifer systems are probably wide spread and have a significant influence on regional recharge. The geochemical composition of groundwater remains relatively consistent throughout the island, with the only irregularities attributed to old groundwater stored within coastal peat.

The outcomes of this study enhance the understanding of groundwater flow, recharge diversion and inhibition for large coastal sand masses in general. For south-east Queensland, it allows the existing regional groundwater flow model to be refined by incorporating independent MRT to test model validities. The location of this large fresh groundwater reservoir, in dry and populous South East Queensland, means its potential to be used as a water source is always high. Background information on aquifer distribution and groundwater MRT are crucial to better validate impact assessment for water abstraction.

# 1   Introduction

Barrier islands are common landforms in coastal environments. They have some of the world's largest biodiversity, provide fresh groundwater resources and have economic value for tourism and mineral industries. The majority of barrier islands





consist of large sand dunes that have high permeabilities. As a consequence, rainfall infiltrates quickly and recharges the aquifer system through a large unsaturated zone (Bryan et al., 2016). Two aquifer systems are commonly present: (i) regional groundwater that is present on the entire island, and (ii) small, localised perched aquifer systems and lakes (usually sealed by the build-up of organic matter or indurated sands) that develop above low permeability layers in the unsaturated zone. Regional groundwater systems develop freshwater lenses beneath these islands until they reach hydrodynamic equilibrium with the underlying saltwater (Röper et al., 2012). The dynamics of these lenses are similar to other coastal aquifers and are affected by changes in groundwater levels due to pumping, land-use change, climate variations, sea-level fluctuations and the hydraulic gradients from the centre of the islands to the coastal discharge areas (Austin et al., 2013; Masterson et al., 2014; Moore et al., 2010; White and Falkland, 2009). Water table fluctuations in the regional aquifer system of the island are mostly caused by the temporal variability of local rainfall, while flow patterns depend on heterogeneity in the aquifer (Austin et al., 2013; Masterson et al., 2014; Moore et al., 2010; White and Falkland, 2009). Variability in the water table of the perched aquifer systems on the other hand results from spatially variable local rainfall and the extent or size of the perched aquifer systems. Although these perched systems are generally small, they are of paramount importance to regional ecology and groundwater dependent ecosystems. They also have the potential to inhibit recharge to the regional groundwater system and to divert the flow pattern of local groundwater.

Many studies exist on the interface of fresh groundwater and seawater in coastal environments (Post et al., 2013a; Mahlknecht et al., 2018; Yechieli et al., 2019; Parizi et al., 2019). The majority of the studies concentrate on either seawater intrusion in coastal aquifers or submarine groundwater discharge (SGD) through the seafloor into the ocean (Stieglitz et al., 2010; Santos et al., 2009; Bryan et al., 2016; Mahmoodzadeh and Karamouz, 2019; Parizi et al., 2019). These studies commonly examine the groundwater-seawater interfaces using hydrodynamic models and hydrogeochemistry rather than estimating recharge rates and water residence times of the terrestrial groundwater on barrier islands.

Although barrier island hydrological systems are reasonable climate indicators, due to substantial variability in their short and long-term climate patterns, current understandings of these systems is generally poor, especially with respect to groundwater (Schneider and Kruse, 2003; Barr et al., 2013, 2019). In particular, due to recharge rates and residence times of water not being sufficiently investigated, the extent of connectivity between regional groundwater and perched aquifer systems is commonly not well understood. Hydraulic connectivity and groundwater flow can be estimated from groundwater bore hydrographs to some extent, but scarce information on unsaturated zone flow and heterogeneities in the aquifer limit the reliability of the outcomes. Flow paths, recharge rates and groundwater residence times can be better constrained by using environmental tracers in combination with aquifer hydraulics. Major ions, trace elements and stable or radioactive isotope tracers, such as $\delta^{18}O$, $\delta^{2}H$, tritium ($^{3}H$ and $^{3}H/^{3}He$) and $^{14}C$ are valuable tools (Voss and Wood, 1994; Han et al., 2012; Röper et al., 2012). CFCs and $SF_6$ have also been used extensively in coastal aquifers (Santoni et al., 2016). In particular, $^{3}H$, $^{14}C$ and CFCs, allow connectivity, recharge rates and most importantly MRT to be assessed.

Increasing water demands from local communities, increased anthropogenic nutrient loading and sea level rises, as well as an increasing periodicity of storm surges due to climate change risk depletion and contamination of the fresh groundwater





bodies. A further consequence is the potential to damage groundwater-dependent ecosystems and destroy a vital groundwater source for coastal regions (Rao and Charette, 2012; Post et al., 2013b; Parizi et al., 2019).

The purpose of this study is to determine groundwater flow paths and MRTs in a Quaternary dune sand aquifer on the world's
second largest sand island, North Stradbroke Island, Queensland, Australia (Laycock, 1975; Ulm et al., 2009). The island is an important part of the coastal environment in South East Queensland and provides a vital groundwater resource, recreational space for the large urban areas and most importantly unique coastal ecosystems with many freshwater wetlands, lakes and marine coastal environments (Marshall et al., 2011). The island also plays a major role in the paleoclimate reconstructions for the east coast of Australia as some of the longest sediment records have been extracted from North Stradbroke Island wetlands
(Barr et al., 2013; Tibby et al., 2016, 2017; Barr et al., 2019). Apart from a small number of government reports (Leach, 2011), there are few studies of the groundwater resource of the island and there is no prior work on residence times. This study closes this knowledge gap using major ion, stable isotope and radioactive tracer data from 21 sites along an east-west transect across the main centre of North Stradbroke Island (Fig. 1). $^3$H and $^{14}$C are used in combination with major ion chemistry and stables isotopes to estimate mean transit times, flow paths of groundwater and potential inter-aquifer mixing on the island.
A more thorough understanding of the islands groundwater flow paths and transit times significantly improves the current understandings of the hydrogeology of North Stradbroke Island and barrier islands in general. This study is relevant to most barrier islands in terms of water resource management and predicting fresh water resources with a changing climate, sea levels changes and increasing urbanisation in coastal environments.

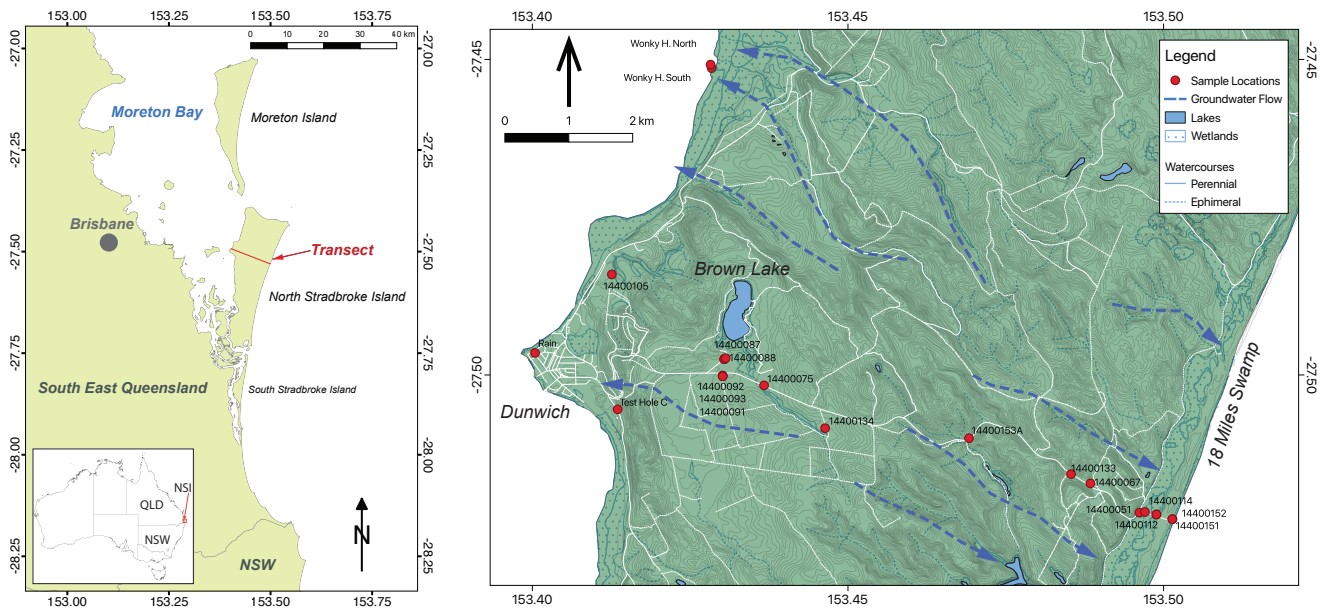

**Figure 1.** Location of North Stradbroke Island in Moreton Bay. Right: Sampled bores by Reference Number (RN) along the transect across the island. Blue arrows indicated general flow directions.



## 2 Geographical, geological and hydrogeological setting

North Stradbroke Island is located approximately 40 km east of Brisbane in a group of sand dune islands that make up Moreton Bay (Fig. 1). The island has an area of approximately 275 km$^2$ of which around 20% is wetlands and beaches with important environmental and economic value. The abundance of fresh groundwater and its proximity to a dry and populous South-East Queensland makes it significant as a water supply. More than 70 wetlands and 25 groundwater dependent ecosystems (wetlands) have been identified on the island. Many of the wetlands date back to the last ice age, which makes North Stradbroke

Island a unique place as a paleoclimate archive (Tibby et al., 2017). It is estimated that 80% of the island will be national park by 2026 (Cox et al., 2013). Most wetlands are internationally important RAMSAR sites that host a large variety of migrating birds and which have important cultural values for the indigenous community (Marshall et al., 2011).

The island consists predominately of large Quaternary on-lapping transgressive parabolic sand dunes, which reach a maximum height of 219 m above sea level, and large low-lying wetland areas around the coast (Cox et al., 2011). There are two

principal land forms, the older and higher Pleistocene sand dunes and the fringing lower Holocene dunes. The dunes overlie Palaeozoic and Mesozoic bedrocks (Rocksburg Greenstone, Woogaroo Subgroup and Triassic Rhyolite) (Kelley and Baker, 1984; Leach, 2011) that are almost completely covered by the dunes except for some small outcrops in the north of the island and around Dunwich (Fig. 1). The largest part of the island is composed of stabilised Pleistocene sand dunes with uniform mineralogy. The sand is well sorted and consists mostly of quartz (∼90-99%) with minor heavy minerals, such as rutile, zircon,

ilmenite, monazite, magnetite and garnet (Laycock, 1975); these have been mined since the 1950's (Moore, 2011).

The climate is subtropical with mean daily temperatures of 15-29 °C in summer and 9-20 °C in winter (Australian Bureau of Meteorology, 2017). Average annual rainfall ranges across the island from 1645 to 1677 mm, with annual evapotransporation ranging from 1095 to ∼1500 mm (Cox et al., 2011). Recharge estimates for the island are derived from water balance models that include climatic factors and groundwater level fluctuations. Long term average estimates are approximately 127 million m$^3$

a$^{-1}$ (1889-2011) in response to the variation in rainfall at a total estimated water volume of the groundwater mound above sea level is approximately 930 million m$^3$ in the regional aquifer. Of the water that leaves the island approximately 16% is through groundwater extraction by local communities and sand mining operations, 55% is discharged to wetlands, 2% is discharged to Blue Lake, and 27% is via submarine discharge at the coastline (Leach, 2011).

The island's groundwater system is predominately hosted in the highly permeable, well-sorted quartz Pleistocene and

Holocene sand aquifer, constrained by the relatively impermeable underlying bedrock. Peat layers and indurated sands, cemented by iron oxides and hydroxides, are widely distributed along the wetlands and swamps. They are considered to control the regional groundwater flow from the elevated sand dunes in the predominantly unsaturated zone (Leach, 2011). The lower permeability units of indurated sand, peat or sandrock/humicrete (locally known as coffee rock) as well as clay accumulations from the weathering of the small amounts of feldspar minerals form widely spread perched aquifer systems. The low perme-

ability layers of the perched systems are believed to originate from chemical leaching of older dune systems in the highlands, and inclusions of marine muds and peats in the lowlands (Laycock, 1975; Thompson and Ward, 1975; Brooke et al., 2008). Some of the perched aquifers and wetland systems are disconnected from the regional aquifer and their locations, but their



extents and genesis are generally poorly understood with only few studies describing the governing processes (Leach, 2011; Barr et al., 2017; Cadd et al., 2018).

The regional groundwater system (which contains almost all of the island's freshwater) forms an elongated mound in north-south direction with the height of the water table varying from a maximum of 42 m AHD (Australian Height Datum) near Mt Hargrave in the centre of the island to surface levels in the coastal lowlands (Cox et al., 2011). Generally, groundwater flows from the high central recharge area, east and west to the coast with a larger proportion moving east due to the relatively elevated bedrock in the west (Fig. 1). Local groundwater discharge points (termed 'Wonky Holes') occur in the tidal areas on the island.

They are round ∼2-10 m diameter disruption features in the tidal muds where groundwater from the regional groundwater system discharges. It is believed that groundwater is semi-confined in paleo-channels in the sand dune system that are covered by fine muds in around Moreton Bay (Stieglitz et al., 2010).

## 3 Materials and Methods

### 3.1 Field sampling and analytical techniques

Groundwater samples were collected from 18 monitoring bores along an east-west transect across the central part of the island (Fig. 1). A number of the sample locations had nested Queensland State Government groundwater monitoring bores sampling both relatively shallow perched aquifer systems and the deeper regional groundwater. The screen depths of the sampled bores vary across the island from 4.6 m below ground level (bgl) lowlands to 131 m bgl in the high central dunes (Tab. A1). A single surface water sample was collected from Eighteen Mile Swamp and two groundwater samples were collected from two Wonky

Holes in the tidal zone approximately 6 km north of Dunwich (Fig. 1). The Wonky Holes were approximately 120 m apart and 6 m in diameter.

  The groundwater bores were sampled in November 2014 while the Wonky Holes were sampled in March 2015. Depth to the water table was determined using an electric water level tape and hydraulic heads were calculated from the surveyed ground elevation of each bore taken from the Queensland Groundwater Database Queensland-Government-Data (2018). Groundwater

was sampled using a Grundfoss MT1 pump where the depth to the screen exceeded 20 m while a smaller 12 V electrical impellor pump (ThemoFisher Inc. Super Twister) was used for the shallower bores. The samples were collected after purging the bores for approximately 5 bore volumes. Bore RN14400075 was pumped dry and sampled using a bailer a day later. In nearly all cases the high hydraulic conductivities of the sands resulted in no or only minor drawdown during purging. At each bore, 6 x1000 mL samples were collected in high density polyethylene (HDPE) bottles, 4 of which were bottom filled and

sealed for $^3$H and $^{14}$C analysis. In-situ measurements of temperature, pH, ORP, DO and EC were taken using a GPS AquaRead with an AP-800 probe (ThermoFisher Inc.). Samples from the Wonky Holes were taken in the middle of the holes by pushing an aluminium tube approximately 1 m into the sand to avoid sampling a mix of fresh water and ocean water. Water was then pumped from the tube using a GeopumpTM peristaltic pump (Geotech Environmental Equipment, Inc.). At the end of each sampling day, water samples were titrated for $CO_2$ and $HCO_3$ concentrations using a Hach titration kit; precision of these

concentrations is 5%. 125 ml of filtered (0.45 $\mu$m) samples were taken for stable isotope and major ion anion analysis. 125





ml of sample was separated, filtered and acidified with 70% $HNO_3$ for later cation analysis. All samples were kept cool until analysis.

The filtered and acidified cation samples were analysed using a ThermoFinnigan quadropole ICP-MS at Monash University. Anion samples were filtered using 0.45 $\mu$m cellulose nitrate filters and concentrations were determined using a Metrohm ion

chromatograph also at Monash University. The precision of major ion concentrations based on replicate analysis is 2-5%. Stable isotopes were measured using Finnigan MAT 252 and DeltaPlus Advantage mass spectrometers at Monash University as described by Hofmann and Cartwright (2013). Precision based on replicate analysis is 0.15‰ for $\delta^{18}O$ and 1‰ for $\delta^2H$. Rainfall stable isotope ratios were taken from the Global Network of Isotopes in Precipitation (IAEA - Global Network of Isotopes in Pricipitation, 2017). Average rainfall weighted isotope ratios for $\delta^{18}O$ and $\delta^2H$ were calculated using $\delta^{18}O$ and

$\delta^2H$ ratios and the daily rainfall from the 1960's to 2002. $\delta^{13}C$ values of dissolved inorganic carbon (DIC) were analysed with a Thermo Delta V continuous flow isotope ratio mass spectrometer, (CF-IRMS) coupled to a Gas Bench II at the University of Queensland. $\delta^{13}C$-DIC values were normalised to the V-PDB scale using international standards NBS19 and LSVEC via two-point normalisation and the precision is 0.3‰.

The Australian Nuclear Science and Technology Organisation (ANSTO) conducted the analysis of the $^{14}C$ and $^3H$ samples

at Lucas Heights, Sydney. $^3H$ was analysed via vacuum distillation, enriched by electrolysis, distilled further to separate tritium and counted using 3 Quantulus ultra-low background liquid scintillation counters (LSC). This analysis had a combined standard uncertainty of 0.04 TU and a quantification limit of 0.05 TU, further analytical details are described in Neklapilova (2008). Following acid extraction and graphitisation of inorganic carbon, $^{14}C$ concentrations were analysed using accelerator mass spectrometry (2MV STAR Tandem Accelerator). The concentrations for $^{14}C$ are expressed as percent modern carbon (pMC)

and the precision of $^{14}C/^{12}C$ ratios is +/- 0.05%.

## 3.2 Estimating MRT

Tritium is a suitable tracer for determining residence times of young groundwater. It is part of the water molecule and once isolated from the atmosphere its activities are only affected by radioactive decay. $^3H$ has a half-life of 12.32 years and may be used to estimate residence times of water that are up to 100 years with a precision of a few years (Morgenstern and Taylor,

2009; Morgenstern et al., 2010). The activities of $^3H$ in rainfall is known with sufficient precision over time in many areas of the world to derive a local $^3H$ input function (Tadros et al., 2014). Brisbane is the closest $^3H$ monitoring station to Stradbroke Island (approximately 35 km from the island) and has a continuous record of $^3H$ data from the 1960s until 2012 (Tadros et al., 2014; IAEA - Global Network of Isotopes in Pricipitation, 2017).

The $^3H$ activities peak in the 1950's to 1960's due to the production of $^3H$ by atmospheric nuclear tests (the 'bomb pulse' $^3H$).

Traditionally, the propagation of the bomb pulse has been utilised to trace the flow of water recharged during this period (Fritz et al., 1991; Clark and Fritz, 1997). Because the bomb pulse $^3H$ peak was several orders of magnitude lower in the southern hemisphere than in the northern hemisphere, $^3H$ concentrations of remnant bomb pulse water in the southern hemisphere have now decayed well below that of modern rainfall. This situation allows estimates of MRT to be obtained from single $^3H$ activities (Morgenstern et al., 2010; Morgenstern and Daughney, 2012).





$^{14}$C concentrations may be used to estimate groundwater residence times in the range 1000 to ~30000 years (Clark and Fritz, 1997). While it is not particularly well suited for estimating residence times of young groundwater it may be used to detect the input of groundwater from old water stores, such as low permeability layers, within the aquifer system (Hofmann and Cartwright, 2013). The combined use of $^3$H and $^{14}$C may also be used to assess mixing of older groundwater with recent recharge (Cartwright et al., 2007). Groundwater MRT were estimated using lumped parameter models (LPM) (Maloszewski

and Zuber, 1982) (Maloszewski, 2000). These models predict the distribution of ages and tracer concentrations in homogenous aquifers with simplified geometries under steady state conditions. The mean residence time represents the average age of the individual water molecules in the sample. The concentration of a radioactive tracer in the sample $C_{out}$(t) is related to the input over time $C_{inp}$(t) via the convolution integral:

$$C_{out}(t) = \int_{et}^{t} C_{inp}(t-\tau)g(\tau)exp(-\lambda\tau)dt \qquad (1)$$

where $\lambda$ is the decay constant of a radioactive tracer (Farlin and Maloszewski, 2013) and g($\tau$) is the transfer function that describes the distribution of ages within the flow system. The piston-flow model (PFM), dispersion model (DM), the partial exponential flow model (PEM), the exponential mixing model (EMM) and the exponential flow model (EPM) are the most commonly used LPMs. The piston flow model assumes that no hydrodynamic dispersion occurs between recharge and discharge area and the MRT calculation is similar to the decay function except that the initial rainfall activity can be varied over

time (Howcroft et al., 2017). The dispersion model is derived from the one-dimensional advection dispersion transport equation and simulates the distribution of a wide variety of aquifer geometries (Cartwright et al., 2017). It allows for variable degrees of dispersion by adjusting the dispersion parameter Dp, which describes the ratio of dispersion to advection. It approaches zero when advection becomes the dominant process controlling the tracer transport. The EMM is most suitable for homogeneous, unconfined aquifers of constant thickness with uniform recharge. The EPM applies to aquifers that have regions of confined

and unconfined flow (Maloszewski and Zuber, 1982; Zuber et al., 2005; Jurgens et al., 2012; Atkinson et al., 2014; Cartwright et al., 2017). The PEM is applicable to the same type of aquifer as the EMM but is used when only the lower part of the aquifer is sampled by a well (Jurgens et al., 2012). The PEM ratio is defined as the ratio of unsampled thickness of the aquifer to the sampled thickness. For bores screened across the total saturated thickness of the aquifer the PEM ratio equals 0 and the PEM equals the EMM.

For this study lumped parameter models contained within the programmable Excel spreadsheet TracerLPM was used (Jurgens et al., 2012). $^3$H activities in rainfall between 1962 and 2012 are from the International Atomic Energy Agency (IAEA - Global Network of Isotopes in Pricipitation, 2017) and Tadros et al. (2014) with data interpolated for missing years were used as the input function. The highest $^3$H activity in 1969 was 84 TU and the $^3$H activity of post 2005 rainfall is 1.6-2.0 TU. It is assumed that pre-bomb pulse rainfall had similar $^3$H activities. The TracerLPM integrated rainfall input function for the

Southern Hemisphere was used for the $^{14}$C activities of local rainfall in this study. The function uses the Southern Hemisphere calibration curve SHcal04 and modern tropospheric $^{14}$C data (Jurgens et al., 2012). The modern data acquisition began in 1955





until 2001 and more modern values are estimated by least squares regression in TracerLPM. Southern Hemisphere long-term input estimates extend to 11,000 years before 1955 compared to the Northern Hemisphere, which extend back to 50,000 years BP. As the $^{14}$C activities in the Southern Hemisphere follow closely those from the Northern Hemisphere, a scaling factor

is applied to the Northern Hemisphere records to estimate Southern Hemisphere input function (Jurgens et al., 2012). MRT were estimated by matching the measured radioactive concentrations to those predicted from the lumped parameter models. While groundwater transit times through aquifer systems are expected to be orders of magnitude larger than short-term rainfall variability, tracer concentrations from accumulated yearly rainfall are the best representation of the tracer input concentration into the system (Hofmann et al., 2018). As the sand aquifer is unconfined across most of the island and bore screens sample

only a part of the aquifer, the partial exponential piston flow model (PEM) with PEM ratios calculated individually for all bores is the best representation for the system. The PEM was compared to the DM and EPM to demonstrate the effect of choice of model to MRT estimations.

## 4  Results

### 4.1  Groundwater hydraulic heads and flow

Groundwater bore 14400088 is probably screened in a perched aquifer system that surrounds Brown Lake (Fig. 1) while all other bores are screened within the regional groundwater system. The distinction between perched and regional aquifer system (in this case regional refers to the connected aquifer system across the island) is made on the bases of nested bores 14400088 and 14400087 in this study. On the day of sampling, the hydraulic head in bore 14400088 was 58 m while that in bore 14400087 was 35 m. This 20 m hydraulic head difference indicates the presence of a perching layer. Bores 14400151 and 14400152 are

located in the coastal beach/shore dunes between the Eighteen Mile Swamp wetland and the ocean on the eastern side of the island.

Hydraulic heads in the regional aquifer are highest (∼35 m) in the centre of the island and the unsaturated zone is approximately 30 to 60 m thick. The hydraulic head values decline towards the coasts and reach sea level in Eighteen Mile Swamp in the east and close to sea level on the western side nearby Dunwich. Long-term decadal fluctuations of regional groundwater

heads are <0.5 m to approximately 5 m. These are not correlated to yearly rainfall and therefore represent longer timescale climate fluctuations (Fig. 2).

### 4.2  Major ion chemistry and stable isotopes

Electrical conductivity (EC) of the groundwater is generally low across the island, ranging from 57 to 257 $\mu$S cm$^{-1}$ with an average of 123 $\mu$S cm$^{-1}$ and TDS values range from 44 mg L$^{-1}$ to 174 mg L$^{-1}$ with an average of 91 mg L$^{-1}$ (Tab. A1). These

values are similar to the average TDS value (78.1 mg L$^{-1}$) across the remaining parts of the island (Queensland-Government-Data, 2018). Higher salinities generally occur closer to the coast while the freshest groundwater is found in the central parts of the island. The high hydraulic gradient from the centre of the island towards the coastal areas inhibits an extensive salt water

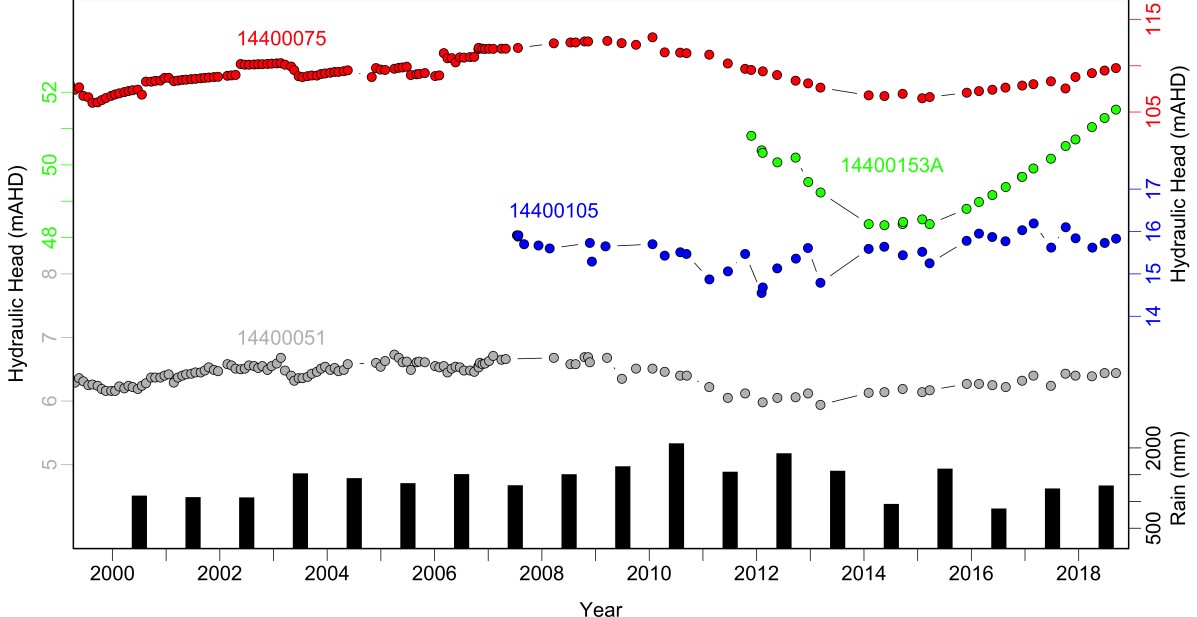

**Figure 2.** Hydraulic heads in meters Australian Height Datum for 4 selected groundwater bores, 14400051, 14400075, 14400105 and 14400153A and yearly rainfall for the period from 2000 to 2018.

wedge developing underneath the island and only shallow areas in beach dunes and intertidal areas have marine dominated groundwater. Most of the groundwater is acidic due to the limited buffering capacity of the relatively clean quartz sands with pH
values ranging from 3.6 to 7.5 with an average of 4.9. Most of the groundwater is oxidised with an average oxidation-reduction potential of +111 mV.

In general, the geochemistry of the groundwater shows only minor variations. Most of the samples (74%) are Na-Cl type groundwater, 21% are Ca-Na-HCO$_3$ groundwater and 5% are Ca-HCO$_3$ groundwater (Fig. A1). Na concentrations range from 9.48 to 29.76 mg L$^{-1}$, Ca concentrations from 0.07 to 14.1 mg L$^{-1}$, Mg concentrations range from 0.88 to 5.82 mg L$^{-1}$, and
K concentrations are generally below 1.2 mg L$^{-1}$. Cl concentrations range from 15.76 to 46.33 mg L$^{-1}$, HCO$_3$ concentrations range from 0.40 to 47.70 mg L$^{-1}$, SO$_4$ concentrations range from 0.15 to 21.76 mg L$^{-1}$ and NO$_3$ concentrations are < 0.5 mg L$^{-1}$ in most groundwater, respectively (Tab. A1). Groundwater from the two Wonky Holes has a similar major ion composition to the inland groundwater. EC values are 98 and 121 $\mu$S cm$^{-1}$ with pH values of 6.79 to 7.00 (Tab. A1). Molar Na/Cl ratios are close to those of seawater (0.86) with a maximum value of 1.2. Mean molar Cl/Br ratios are close to the average Cl/Br ratio
of ocean water and coastal precipitation of ∼650 (Davis et al., 1998) (Fig. 3 A). Ca/HCO$_3$ ratios range from 0.01 to 0.96 with higher ratios generally occurring towards the coasts.

The $\delta^{18}$O and $\delta^2$H values of inland groundwater have a small range from -5.4‰ to -2.4‰ and -32.3‰ to -24.4‰, respectively (Tab. A1). The one rainfall sample has $\delta^{18}$O and $\delta^2$H values of -3.2‰ and -25‰, slightly higher than those of average

**Figure 3.** Geochemistry of surface water and groundwater from the bores and Wonky Holes. A) Na vs. Cl, B) Ca vs. Cl, C) Na/Cl vs. Cl, D) Cl/Br vs. Cl, E) Mg vs. Cl, F) Mg vs. SO₄, G) Ca vs. HCO₃ and H) NO₃ vs. TDS.



rainfall for Brisbane ($\delta^{18}$O = -3.98‰, $\delta^2$H = -18.4‰, (Hollins et al., 2018). The $\delta^{18}$O and $\delta^2$H values of both wonky hole

samples are -4.6‰ and -23‰, which are within the range of the inland groundwater. All waters plot close to the Brisbane

meteoric water line (Crosbie et al., 2012; IAEA - Global Network of Isotopes in Pricipitation, 2017; Hollins et al., 2018) (Fig.

4).

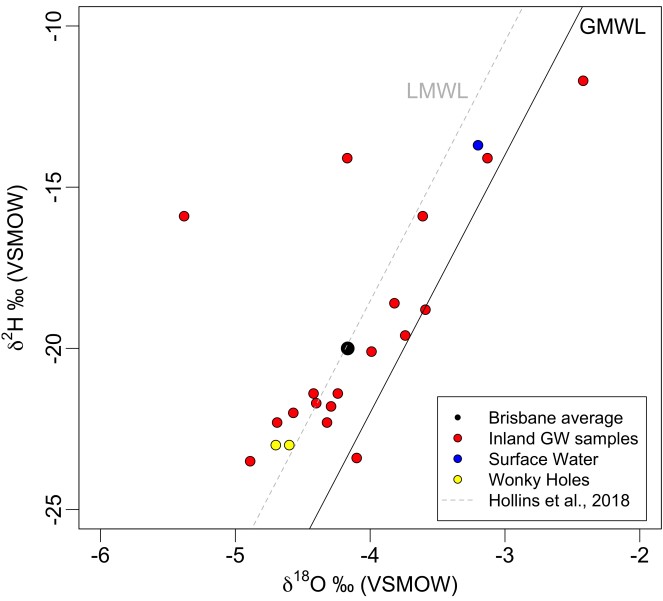

**Figure 4.** $\delta^{18}$O and $\delta^2$H values of groundwater, surface water and water from the Wonky Holes on North Stradbroke Island. The black point represents the rainfall weighted average composition of $\delta^{18}$O and $\delta^2$H for rainfall from the Global Network of Isotopes in Precipitation for Brisbane from 1962 to 2002 (Hollins et al., 2018).

### 4.3    $^3$H, $^{14}$C and $^{13}$C

$^3$H activities of groundwater range from below detection (< 0.05 TU) to 1 TU (Tab. A1). They are much lower than mean

annual $^3$H activities in modern precipitation in Brisbane of approximately 1.6 - 2.0 TU (Tadros et al., 2014). $^3$H activities

generally decrease with bore depth and increase with distance from centre of the island towards the coastlines (Fig. 5 B). The

decrease is more pronounced towards the east coast then towards the west coast. Groundwater from bores 14400134, which is

the deepest bore at 131 m below natural surface in the central part of the island, and bore Testhole C, which is located close

to the west coast at a depth of 43 m (Fig. 1), have $^3$H activities that are below detection. The water discharging from the two

Wonky Holes also has low $^3$H activities of 0.12 and 0.15 TU. The highest $^3$H activity of 1 TU is from groundwater in bore

14400088, which is a shallow bore (4.6 m) in a perched aquifer system close to Brown Lake.

Most $^{14}$C concentrations range from 59 to 111 pMC with the majority of samples having concentrations above 90 pMC.

The lowest $^{14}$C concentrations are in groundwater from bores 14400134 (77 pMC) and Test hole C (63 pMC) (Fig. 6 A &





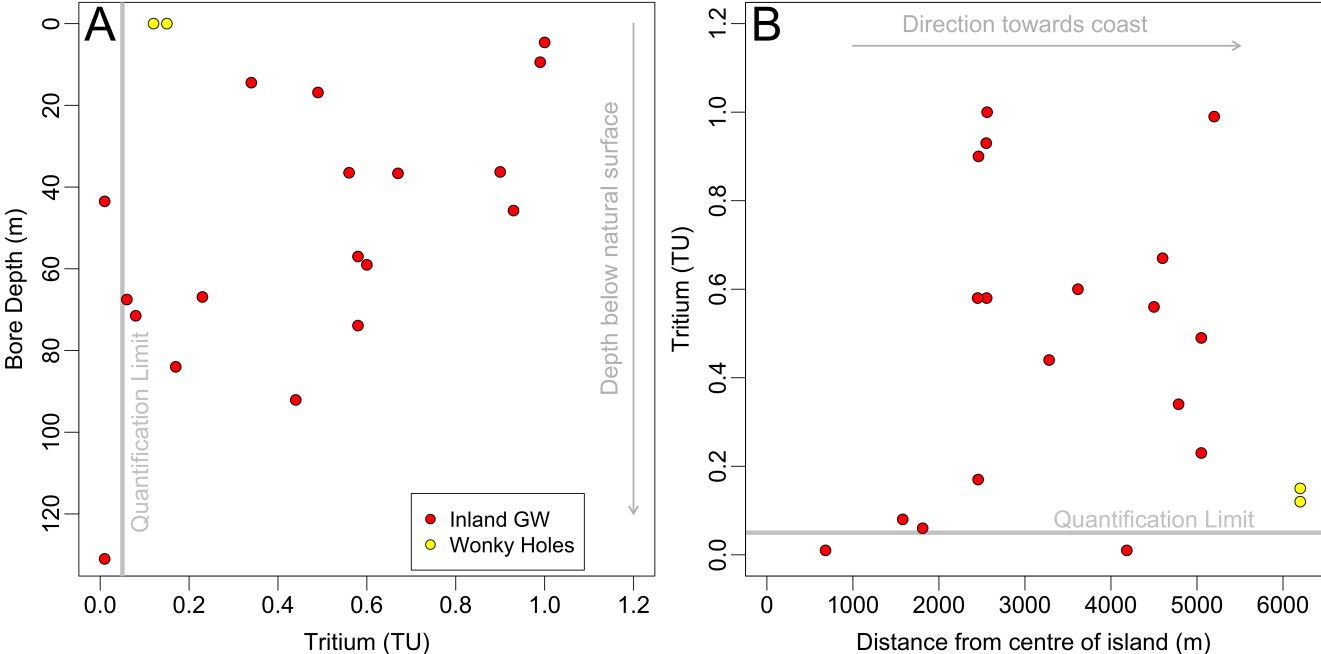

**Figure 5.** A) Variation in $^3$H activities with bore depth, B) $^3$H activities vs. distance from the centre of the island, which is approximately the groundwater flow divide between west- and eastwards flow.

B), which also had the lowest $^3$H activities. Groundwater from the Wonky Holes has $^{14}$C concentrations of 81 and 59 pMC.

Generally, the high $^3$H activities and high $^{14}$C concentrations of most of the groundwater imply that it was recently recharged.

$\delta^{13}$C values of DIC (Fig. 6) range from -25‰ to -10.7‰. Most of the waters have $\delta^{13}$C values in the range of -24‰ to -18‰, which are within the expected range of $\delta^{13}$C values of DIC derived from dissolution of soil $CO_2$ in an environment dominated by C3 vegetation (Clark and Fritz, 1997). The highest $\delta^{13}$C values are from groundwater from bores 14400112 (-10.6‰) and 14400152 (-10.7‰) on the east side of the island at Eighteen Mile Swamp and the beach dunes. Groundwater from the shallow

bore in the perched aquifer system, bore 14400088, has the lowest $\delta^{13}$C value of -25‰. The Wonky Holes both have $\delta^{13}$C values of -12‰ (Fig. 6 A).

## 5 Discussion

The combined hydraulic head and geochemistry data allow the conceptualisation of groundwater flow across North Stradbroke Island. Groundwater flows from the centre of the island towards the east and the west coasts driven by the large hydraulic

gradient. There is groundwater discharge into freshwater wetlands (e.g. Eighteen Mile Swamp), some submarine groundwater discharge via the Wonky Holes into Moreton Bay and probably offshore on the eastern side of the island (Fig. 1). The sand dunes are thickest in the central dune field and the thickness gradually declines towards both coastlines. The thinning of the





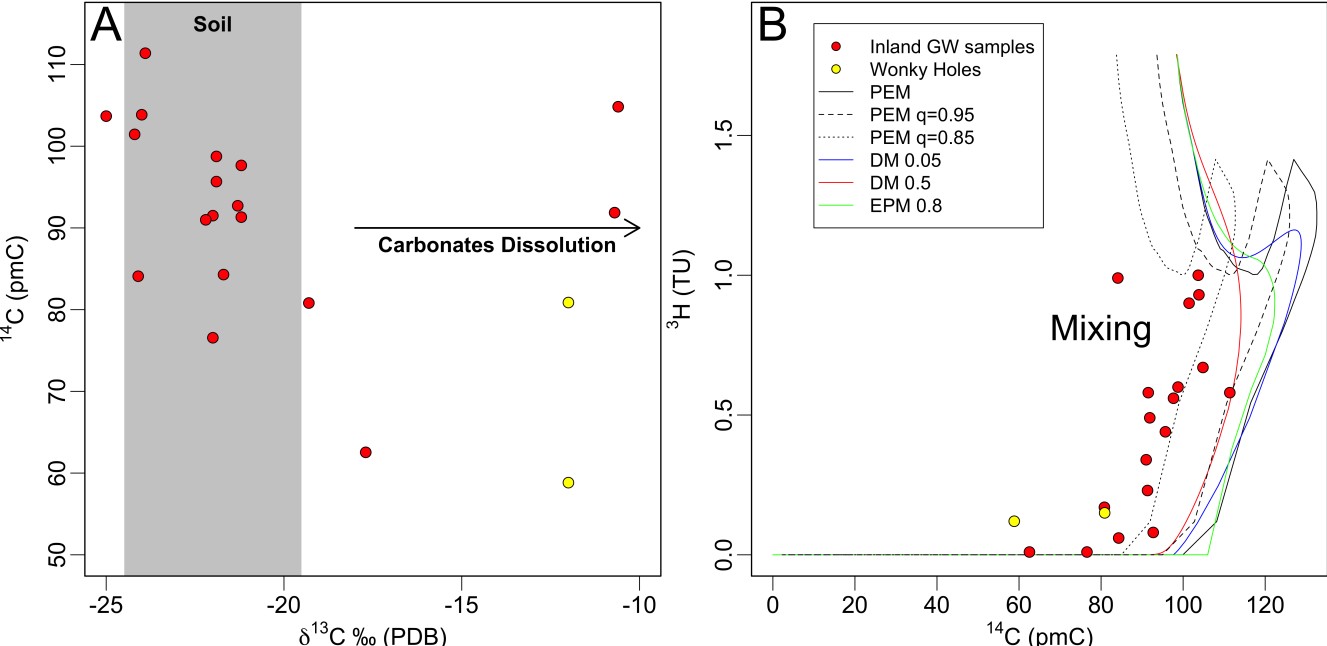

**Figure 6.** A) $^{14}$C concentration vs. $\delta^{13}$C values of groundwater and water from the Wonky Holes. The shaded grey area represents the predicted range of $\delta^{13}$C values of DIC in the soils. B) $^{14}$C concentrations vs $^{3}$H activities of groundwater and the water from the Wonky Holes. The curves show the predicted covariance from the lumped parameter models (PEM = Partial Exponential Model with dilution factors q =0.95 and q=0.85, DM = Dispersion Model (with dispersion parameter 0.05 and 0.5), EPM = Exponential Piston Flow Model (with EPM factor of 0.8))

aquifer and the unsaturated zone allows mixing of groundwater from the centre of the island with more recent recharge closer to the coast towards the discharge zones.

Most of the groundwater has low TDS and low pH. This together with the low Na/Cl ratios ($< 1.2$) suggests only minor silicate weathering is occurring, limited by the availability of weatherable silicates. The linear relationship between Ca and HCO$_3$ (Fig. 3 G) suggests that some carbonate weathering has occurred. However, the observation that $\delta^{13}$C values are generally similar to those expected for DIC derived from the soil zone implies that this is limited. SO$_4$ and Cl concentrations increase near the coastline suggesting some marine influence (sea spray) in the coastal lowlands of the island. However, the salinity remains low, supporting the argument that the salt water wedge underneath the island is relatively deep close to the bedrock.

## 5.1 Mean groundwater residence times

The choice of the best fit lumped parameter model requires conceptualisation of the flow system. The exponential-piston flow model is commonly used to calculate MRT in flow systems that have near-vertical recharge through the unsaturated zone overlying a flow system with an approximately exponential distribution of flow paths (Cook and Simmons, 2000). However,





the exponential-piston flow model assumes that the entire aquifer thickness is sampled, which is not the case. Here we use the partial exponential model and calculate the PEM ratio for each bore using the screen width and depth below the water table. This model assumes that flow system is exponential and so ignores the likelihood of piston flow in the unsaturated zone. However, as the thickest unsaturated zone is approximately 60 m and the total flow path length is ∼5000 m, the proportion of the piston flow component is very small.

MRT calculated using the partial exponential flow model range from 37 years to >150 years. Below background $^3$H activities in groundwater samples from Test Hole C and 14400134 suggest MRT past the age range of tritium (Tab. A2). Deep groundwater in the centre of the island where the sand dunes overlay the bedrock has MRT > 150 years (Fig. 8 B). The MRT in the eastern part of the island are generally younger (50-124 years) than in the western part (37- >150 years). There are several uncertainties in estimating MRT. Varying the $^3$H activity of modern rainfall between 1.6 TU to 2.0 TU have little effect on the 305 MRT estimate and lie at less than 1% for the MRTs derived by the dispersion model and there is no difference for MRTs using the other models. Macroscopic mixing within aquifers (aggregation) may affect MRT (Stewart et al., 2017) but mainly where bimodal mixing of young and old waters occurs. The more complex mixing of water of different MRT in aquifers reduces the error associated with aggregation as this is similar to the flow in the lumped parameter models (Cartwright and Morgenstern, 2016). The main uncertainty in the MRT calculations, especially for older waters, is the choice of lumped parameter model 310 (Cartwright et al., 2018). While the partial exponential model accords with the conceptualisation of the flow system, it will not represent it in detail. To demonstrate the effect of LPM choice, MRT were also calculated using the dispersion model with dispersion parameters 0.05 (mostly advection) and 0.5, as well as an exponential piston flow model with piston flow to exponential flow ratio of 0.5. The difference in MRT between the models is low when $^3$H activities are high (>1 TU) but increase markedly as $^3$H activities decrease. For $^3$H activities <0.2 TU, the range of MRT is 75-290 years, indicating that MRT cannot 315 be reliably calculated. Additionally, low $^3$H waters are more susceptible to contamination during sampling or analysis. While determining MRT is subject to uncertainties, the relative distribution of older and younger water does not change according to which lumped parameter model is used (Fig. 7 A).

While groundwater was expected to have young MRT some of the groundwater has relatively low $^{14}$C activities. Estimating MRT using $^{14}$C activities requires that the addition of $^{14}$C-free carbon from the groundwater flow system be accounted for. 320 Significant addition of $^{14}$C-free carbon may dilute the $^{14}$C activities, potentially resulting in MRT being overestimated (Coetsiers and Walraevens, 2009). Major ion geochemistry and the $\delta^{13}$C values of DIC indicate minor calcite dissolution in some of the groundwater. The proportion of $^{14}$C derived from recharge (q) is calculated from a $\delta^{13}$C mass balance using the measured $\delta^{13}$C of the groundwater samples and estimated $\delta^{13}$C value for DIC in recharge and carbonates (Clark and Fritz, 1997; Cartwright et al., 2017; Hofmann and Cartwright, 2013). Most of the $\delta^{13}$C values from the sampled groundwater are close to 325 those for DIC derived from the local soils and q values are close to 0.95 (range of 0.61-1.03, median=0.95). These q values are higher than those generally proposed for sediments containing fine-grained carbonates (0.75-0.9) and more similar to those in silicate-dominated crystalline rocks (0.9-1.0) Clark and Fritz (1997). They are consistent, however, with the sands being silica rich and the limited carbonate dissolution implied by the geochemistry. The Ca and HCO$_3$ concentrations in groundwater are also similar to those in groundwater from in crystalline rocks (Tweed et al., 2005; Le Gal La Salle et al., 2001; Coetsiers and



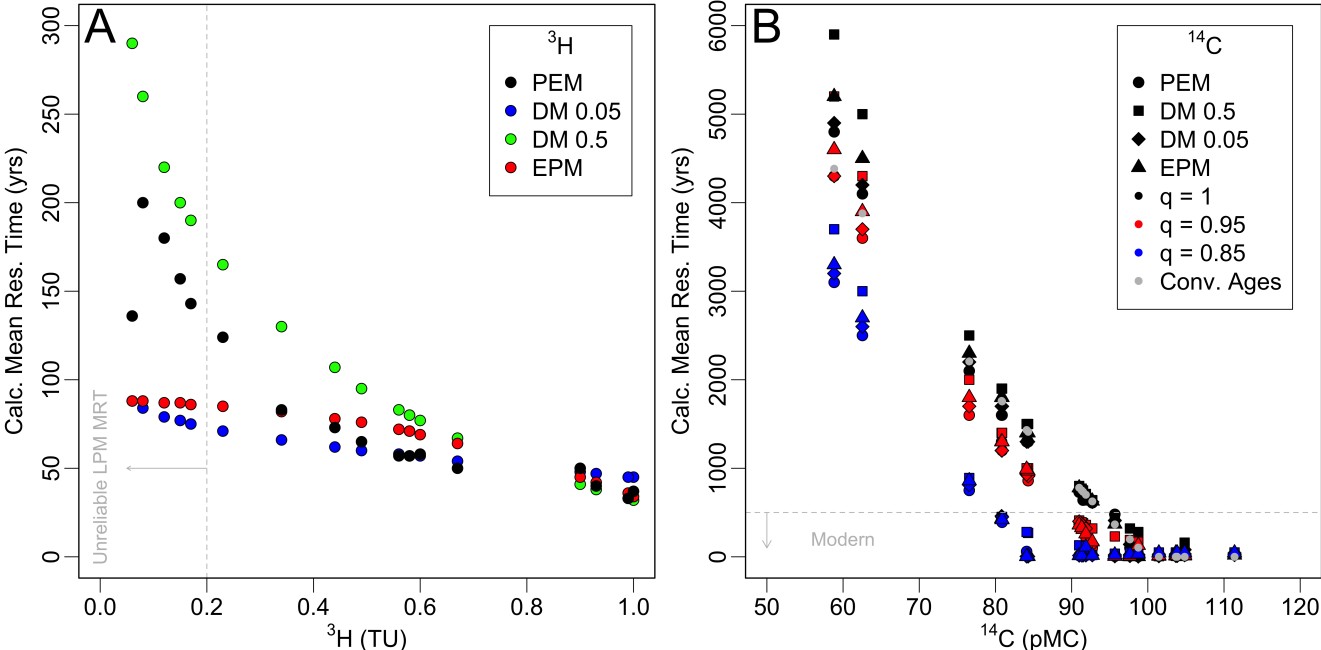

**Figure 7.** A) Comparison of calculated MRT for the Partial Exponential Model (PEM), the Dispersion Models (DM with dispersion parameters 0.05 and 0.5) and the Exponential Piston Flow Model (EPM) using $^{3}$H. B) Comparison of calculated MRT for the Partial Exponential Model (PEM), the Dispersion Models (DM with dispersion parameters 0.5 and 0.05) and conventionally calculated carbon ages using the decay function. The model MRT were also calculated using $^{14}$C dilution models with q factors of 0.95 and 0.85.

Walraevens, 2009). Groundwater from bores 14400152, 14400112, Test Hole C and the Wonky Holes, have higher $\delta^{13}$C values and Ca and HCO$_3$ concentrations and may record a higher degree of calcite dissolution.

In addition to calcite dissolution, microbial degradation of organic matter in the aquifer or the very low permeability sediments around the lakes and wetlands may contribute $^{14}$C-free carbon. (Tab. A2). Most of the deeper groundwater has little dissolved organic carbon (DOC) (not measured but colour indicated DOC or the lack of) but the groundwater recharged through

the peat rich wetlands often contains larger amounts of DOC. The breakdown of organic matter as an origin of $^{14}$C-free carbon is more likely but it would not have occurred in-situ where the samples were taken for a number of reasons; a) most of the waters are slightly oxygenated, b) while there is very little NO$_3$, concentrations of SO$_4$ are relatively high and c) the isotopic shift of $\delta^{13}$C values towards more enriched values would be more pronounced. It seems more likely that organic matter degradation processes occurred in the peat sediments around lakes and wetlands and the seepage from those mixed with existing water in

the sand aquifer. The maximum amount of dilution by $^{14}$C-free carbon derived from calcite dissolution organic matter may be estimated from the covariance of $^{3}$H and $^{14}$C (Fig. 6) (Cartwright et al., 2013). Reducing q values displaces the covariance curves to lower $^{14}$C. While mixing with old groundwater can result in waters lying to the left of the covariance curves, it is

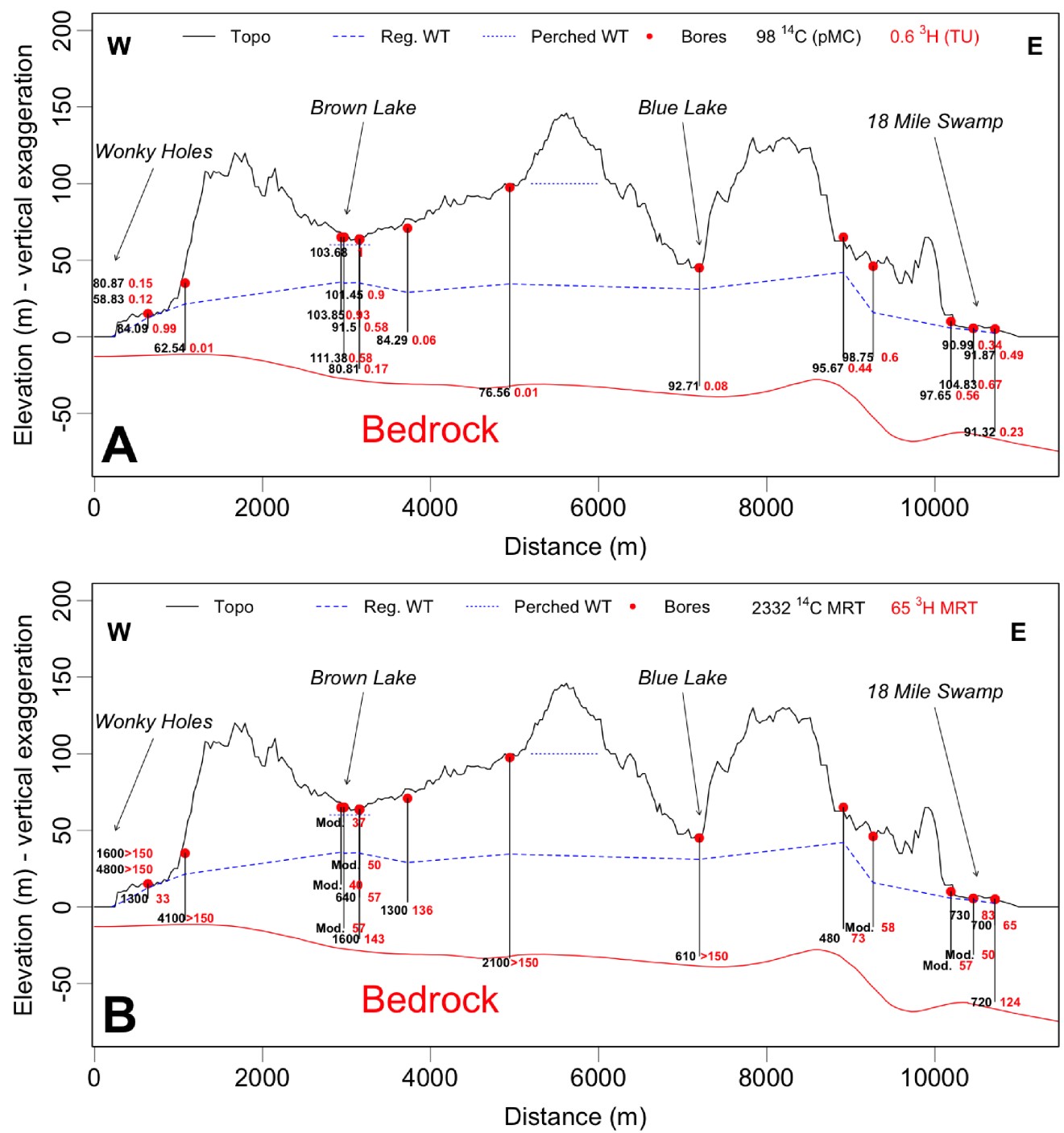

**Figure 8.** A) Vertically exaggerated cross-section along the samples transect with bore locations, depth and pMC (black) as well as tritium activities (red) B) Estimated MRT calculated for tritium activities (rainfall 1.6 TU) with an exponential piston flow model and [14]C ages after correction (q=0.95).





not possible for waters to have higher $^{14}$C as that would require the initial $^{14}$C activity to be greater than that recorded in the atmosphere. In the case of Stradbroke Island groundwater, this implies that q values cannot be substantially lower than 0.8.

MRT were calculated by adjusting the $^{14}$C input function using q values of 0.85-0.95. Some of the adjusted groundwater $^{14}$C MRTs are younger than 200 years and are therefore considered to be modern (Fig. 7 B). MRT calculated using the partial exponential model range from modern to 4800 years. MRT of groundwater are generally higher on the eastern side of the island. The MRT in the Wonky Holes (Wonky Hole South 4800 years, Wonky Hole North 1600 years) and in bores Test Hole C (4100 years), 14400134 (2100 years), 14400092 (1600 years), 14400105 (1300 years) and 1440075 (1300 years) is older than

the majority of groundwater in the main aquifer system. The partial exponential flow model produces the youngest MRT using $^{14}$C (Fig. 7 B; Tab. A2). Differences are more pronounced in the older waters with maximum of 1100 years difference (partial exponential model versus dispersion model with dispersion parameter 0.5) but relative MRT distributions across the models remain the same. Applying the dilution factors of 0.85 and 0.95 to the calculations has a much stronger effect on the MRT. The range of MRT of groundwater using the partial exponential model with the dilution factor 0.85 for example produces a range

of modern to 3100 years where Wonky Hole South, Test Hole C 14400134 and 14400092 have still the largest MRT with 3100, 2500, 750 and 420 years, respectively.

## 5.2   Disparities in groundwater MRT

The calculated groundwater MRT from the radioisotope tracers are generally higher than what was initially hypothesised. Existing MRT were based on flow rates estimated using the regional groundwater model Leach and Gallagher (2013). This

assumed lateral hydraulic conductivities of 1 to 40 m day$^{-1}$ based on Leach and Gallagher (2013). The large variance in hydraulic conductivities comes from the inclusions of isolated peat and clay layers (Leach and Gallagher, 2013). By contrast hydraulic conductivities calculated via Darcy's Law with velocities from the $^{14}$C MRT, porosities from Leach (2011) and the measured hydraulic gradients are generally between 0.25 and 1 m day$^{-1}$. There are two likely explanations for these lower hydraulic conductivities; a) some groundwater discharge to the unconfined sand aquifer from the basement units underneath

the sand dunes, and b) a larger volume of geological units with lower permeability such as peat, coffee rock and clay with more control on groundwater flow than prior studies suggest.

## 5.3   Potential influence of geological basement

The geological basement underneath the island is comprised of the Woogaroo Subgroup, Rocksberg Greenstone and Rhyolitic intrusions. A regional evaluation of aquifer storage and retention for South-East Queensland identified aquifers within the

units summarising some of the general hydrogeological characteristics, such as permeabilities, transmissivities and general groundwater flow (Helm et al., 2009). In the case of the Rocksberg Greenstone water is stored in fractures infilled with clay and the typical yield is less than 0.2 Ls$^{-1}$. The Woogaroo Subgroup sandstone has a typical yield ranging from < 1.5 Ls$^{-1}$ to 6 Ls$^{-1}$ and was identified to have moderate groundwater storage potential (Neuman, 2005). There is a possibility that the underlying basement units on the island are connected to the lower sand dunes and groundwater enters from these formations

the young dune water. The basement isopach, which was extrapolated from the regional geological model for South-East



Queensland, indicates the proximity of each sampling location to the bedrock (Fig. 8 B). The closer the groundwater samples are to the basement geology the older their respective MRT. This in itself is not a compelling argument, as it is also in line with the progression of MRT from shallow to deeper sections of the aquifer but having larger MRT in the west of the island where the bedrock is shallow might indicate a small degree of discharge from the basement to the dune aquifer. To the best of our knowledge there are no groundwater bores in the basement on the island but general groundwater heads on the mainland suggest that this could be a possibility, however, the volumes would be minor, if at all, compared to the volume of the sand aquifer.

## 5.4    Potential influence of lower permeability units on MRT

Over the course of the Quaternary, accumulation of sand along the coast of south-east Queensland formed the large sand masses in the area. The dune formation came with periods of dune migration and relatively stable periods where dunes were stagnant (Barr et al., 2013; Ellerton et al., 2018). During these periods the decomposition of organic material, weathering of minerals and fluctuations in the water table with subsequent redox reactions have led to the formation of lower permeability units within the otherwise relatively permeable dune systems. The geomorphological, geochemical and environmental processes around the formation of these lower permeability units are not entirely constrained but are often linked to pedogenic processes or the accumulation of fine materials in surface depressions. They led to the occurrence of the aforementioned iron crusts, coffee rocks and thick peat sections in the dune stratigraphy. The iron crusts in particular are believed to be linked to the soil B horizons and are expected to divert groundwater flow in some areas of the island while the coffee rocks and peats are formed around wetland and lakes (Cendón et al., 2014). All these formations have most likely intermediate water storages with much greater MRT than the actual sand aquifer. Permeabilities for the iron crusts are not known and thicknesses are variable but some peat formations are more than 10 m thick and permeabilities in the peat decline rapidly once a certain thickness (below 2-3 m depth) is reached resulting in the storage of water over long residence times (millenial). Slow leakage or seepage, in particular with hydraulic loading under increased recharge events may add water with a very long MRT to the general water pool in the sand aquifer system. While this is a potential cause that can produce a mixed groundwater from sources with very distinctly different residence times, it would most likely occur in the proximity of these peat units. There is proof for this type of scenario when comparing bores 14400114 and 14400112. They are two nested bores on the edge of Eighteen Mile Swamp, a large freshwater peatland on the Eastern side of the island. Bore 14400114 is the shallower bore but has a much older MRT (pMC = 91) than the deeper bore 14400112 (pMC = 105). Comparing major ion and trace element concentrations indicates that the groundwater sample from bore 14400114 has uncharacteristically high concentrations of in Na, Mg, Sr, Si, Mn, Fe, Cl, Br and $HCO_3$. Almendinger and Leete (1998) found elevations in the same major ions and trace elements when comparing wells above peat layers to wells below peat layers. The large enrichment of elements seen in this sample and its disparity to other MRT suggests that this bore receives groundwater from the peat unit forming Eighteen Mile Swamp where the hydraulic conductivities are relatively low and concentration of the aforementioned ions and elements are high.





## 5.5 Conceptualisation of groundwater flow

Variations in groundwater MRT estimated from $^3$H and $^{14}$C throughout North Stradbroke Island support the current idea of

groundwater flow from the centre of the island towards both coasts. There is also robust evidence that groundwater has a large vertical flow component (Fig. 9). Nested bore sites show a distinct increase in MRT as bores protrude deeper, while single bores that extend deeply also display increased MRT (Fig. 8 A & B). While the MRT variation is not as strongly pronounced in lateral flow direction, the combination of known hydraulic gradients and known areas of discharge give prove to the idea of lateral flow to be consistent with MRT distributions. Modelling by Chen et al. (2003) and Leach and Gallagher (2013) suggests that

groundwater on North Stradbroke island is discharged via: a) movement into coastal wetlands (55%), b) submarine discharge around the coastline (27%), c) groundwater extraction (16%) and d) discharge into blue lake (2%) (Chen et al., 2003). The conceptual model of groundwater flow along the sampled transect is divided into two areas. An area where groundwater flows to the west and an area where it flows to the to the east (Fig. 9). There are minor flow diversions to the north and the south resulting from the centre of the transect also being the highest elevation on the island and topography drops from there into all

four directions. It is assumed that recharge occurs across the whole island and as such recent recharge is added to the lateral groundwater flow at all points along the cross section.

The east coast of the island is comprised of a large freshwater wetland system at the foothills of the major dune system. This wetland, Eighteen Mile Swamp, separates the major Quaternary dunes from the fore dunes. Groundwater from the main dunes discharges into the wetland and re-enters partially into the fore dunes along the coastline (Fig. 9). However, it appears that there

are at least two aquifer systems that are separated by a lower permeability unit. This lower permeability unit was identified as a marine clay underneath the peat sequences of Eighteen Mile Swamp by Mettam et al. (2011). Nested bores 14400151 (67 m) and 14400152 (17 m) indicate younger MRT and higher salinity (173 mg L$^{-1}$ TDS) in the shallow part of the fore dune system and older, lower salinity in the deeper parts which is also reflected in bore 14400051 in the main dune system (Fig. 1). There is also an upward gradient indicating a hydraulic connection of the deeper fore dune to the regional groundwater system

and potentially to Eighteen Mile Swamp (Fig. 9).

The west coast of the island has generally lower topographic gradients and extended salt marshes that continue into the mangrove tidal areas of Moreton Bay. Sequences of fine, organic matter rich muds overlay the dune sands which lead to semi-confined conditions in the sand aquifer in the direct proximity of the coast underneath the muds (Fig. 9). The Wonky Holes are circular disturbances in the mud sequences where fresh groundwater from the island's sand dunes discharges into the salt water

environment of the bay. The controls surrounding the formation of these discharge points come to existence is unknown, but an upward head gradient in the underlying confined system, bioturbation and heterogeneities in the mud sediments possibly created an opening to the surface. The long MRT of water discharging through the Wonky Holes suggest that most of the water derives from the deeper confined sand aquifer units that are linked to the centre of the island (Fig. 9).

The results of this study indicate that the perched groundwater systems have a significant effect on groundwater flow,

recharge inhibition and intermediate water storage. Many lake and wetland systems on the island exist around perched aquifer system and are in some instances the cause of the perching layer formation. Depending on age and location of the perched





**Figure 9.** Conceptual sketch of groundwater flow across the island from the centre of the island to the ocean in the east and to Moreton Bay in the west. Parts of the centrally recharged groundwater also flows to the north and thesSouth (Leach and Gallagher, 2013). Lower permeability units in the unsaturated zone can have longer residence times than direct recharge. There also may be groundwater contributions from the bedrock. On the western side some of the groundwater is discharged directly into Moreton Bay, some flows underneath the tidal mud flats in semi-confined conditions and is discharged partly through Wonky Holes into the Bay.





systems, perching layer vary largely in thickness. Permeabilities decrease with thickness and some of the systems may have very slow transit times through the perching layers resulting in groundwater leaking into the main aquifers that has potentially MRT of thousands to tens of thousands of years. Even small amounts of leakage from these systems have the ability to lower

overall groundwater MRT.

## 6   Conclusions

In summary, the combination of cosmogenic isotopes, major ion chemistry and stable isotope geochemistry were used to conceptualise groundwater flow on North Stradbroke Island with varying groundwater MRT in different parts of the island and determine groundwater flow paths through the aquifer systems. The groundwater MRT in this study differ to those found in

similar studies of barrier island groundwater systems (Röper et al., 2012). The reason for the differences is that: the dune stratigraphy of North Stradbroke Island is more heterogeneous with inclusion of peat, clay and indurated sands; volcanic and sedimentary basement units are underneath the island; and the island is much larger with a more variable topography, geomorphology and vegetation. MRT estimated using [3]H indicate a strong vertical stratification from 37 years to >150 years. [14]C MRT display similar temporal relationships with much greater ranges. This MRT discrepancy is attributed to different

groundwater reservoirs. This study did not produce evidence for contributions from the fractured Woogaroo Subgroup sandstone aquifer but the possibility remains. Water diversion and retention by low permeability units in the dune systems are so far the most likely course for relatively long MRT. The geochemical composition of groundwater remains relatively consistent throughout the island, with the only irregularities attributed to old groundwater stored within coastal peat. The stable isotope composition of North Stradbroke Island groundwater is similar to Brisbane precipitation without any indication of evaporative

enrichment. The outcomes of this study can be incorporated in regional groundwater flow models to refine the potential inhibition and retardation of recharge to test model validities. The position of the islands large fresh water reservoir in a dry and populous South East Queensland means its potential to be used as a water resource is always high and background information on aquifer distribution and groundwater MRT are crucial to better validate impact assessment for water abstraction.

*Acknowledgements.*  We want to thank the traditional owners of the land and water around Moreton Bay and North Stradbroke Island (Min-

jerribah), represented by the Quandamooka Yoolooburrabee Aboriginal Corporation (QYAC), for their continuing support and collaboration on their land. We also want to thank the staff of the University of Queensland Moreton Bay Research Station in Dunwich for their continuous support and help in preparing and executing field work. This research project was partially funded by the School of Earth Sciences at the University of Queensland and the School of Earth, Atmosphere and Environment at Monash University. [3]H and [14]C analyses were funded by the Australian Nuclear Science and Technology Organisation (ANSTO), award number ALNGRA14530.



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





**Appendix A**

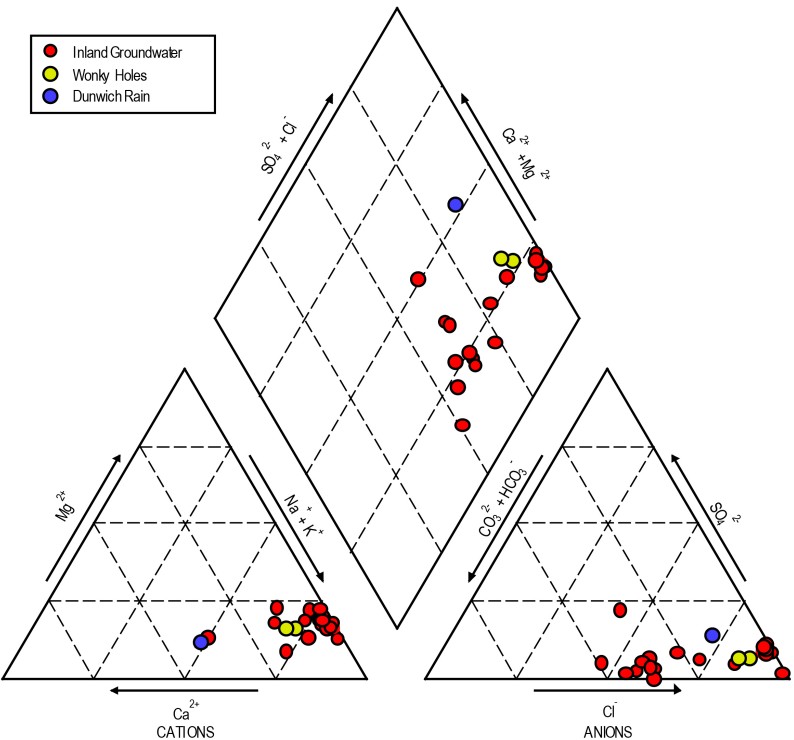

**Figure A1.** Piper diagram showing inland groundwater samples (red), samples from the two Wonky Holes (yellow) and rain water sample (blue).




**Table A1.** Chemistry data, Lat=Latitude, Lng=Longitude, Water Level= water level below natural surface, Hydr. Head=hydraulic head, Major ion data, molar Cl/Br and Na/Cl ratio. CBE=Charge balance error.

| RN | Lat | Lng | Bore depth | Water Level | Hydr. Head | Temp | pH | TDS | DO | ORP | EC | $\delta^{18}O$ | $\delta^{2}H$ |
| --- | --- | --- | --- | --- | --- | --- | --- | --- | --- | --- | --- | --- | --- |
| Units | | | m | m | m | °C | | mg L$^{-1}$ | mg L$^{-1}$ | mv | µS/cm | ‰ | ‰ |
| 14400051 | -27.521831 | 153.496183 | 36.5 | 0.87 | 5.7 | 22.6 | 5.3 | 71 | 1.94 | 206.3 | 124 | -4.9 | -23.5 |
| 14400067 | -27.517191 | 153.488406 | 59.0 | 30.45 | 15.9 | 24.6 | 4.4 | 56 | 2.17 | 238.8 | 94 | -4.7 | -22.3 |
| 14400075 | -27.501655 | 153.436704 | 67.5 | 42.23 | 29.0 | 27.1 | 5.1 | 106 | 0.5 | 98.7 | 104 | -4.6 | -22.0 |
| 14400087 | -27.497494 | 153.430401 | 73.9 | 24.77 | 35.9 | 21.4 | 3.9 | 98 | 1.38 | 187.3 | 67 | -3.7 | -19.6 |
| 14400088 | -27.497503 | 153.430370 | 4.6 | 2.13 | 58.3 | 19.1 | 3.7 | 44 | 1.67 | 74.8 | 77 | -4.2 | -14.1 |
| 14400091 | -27.500095 | 153.430138 | 36.3 | 28.90 | 35.4 | 22.9 | 4.3 | 63 | 0.56 | 224.6 | 57 | -5.4 | -15.9 |
| 14400092 | -27.500158 | 153.430148 | 84.0 | 28.90 | 34.9 | 23.0 | 5.2 | 92 | 1.39 | 1.5 | 105 | -4.4 | -21.7 |
| 14400093 | -27.500231 | 153.430158 | 57.0 | 28.96 | 35.0 | 23.0 | 5.4 | 97 | 1.31 | -19.7 | 105 | -3.8 | -18.6 |
| 14400094 | -27.497349 | 153.430613 | 45.8 | 25.38 | 35.7 | 21.6 | 4.4 | 78 | 1.24 | 35.5 | 66 | -3.6 | -15.9 |
| 14400105 | -27.484070 | 153.412593 | 9.4 | 3.18 | 12.5 | 21.5 | 4.0 | 173 | 0.74 | -7.6 | 150 | -3.6 | -18.8 |
| 14400112 | -27.522101 | 153.498858 | 36.7 | 1.78 | 4.3 | 20.4 | 5.1 | 219 | 2.26 | -52.5 | 257 | -3.1 | -14.1 |
| 14400114 | -27.522119 | 153.498868 | 14.5 | 1.72 | 4.4 | 22.3 | 4.4 | 68 | 2.51 | 206 | 124 | -4.3 | -22.3 |
| 14400133 | -27.515712 | 153.485342 | 92.1 | 36.48 | 42.1 | 22.9 | 4.9 | 50 | 1.85 | 233.5 | 73 | -4.2 | -21.4 |
| 14400134 | -27.508440 | 153.446380 | 131.0 | 63.46 | 34.5 | 24.6 | 4.5 | 67 | 0.84 | 204.9 | 75 | -4.4 | -21.4 |
| 14400151 | -27.522841 | 153.501403 | 66.9 | 3.10 | 2.3 | nd | nd | 59 | nd | nd | nd | -4.1 | -23.4 |
| 14400152 | -27.522850 | 153.501372 | 16.9 | 3.62 | 1.8 | 21.1 | 5.8 | 174 | 2.6 | 62 | 220 | -4.0 | -20.1 |
| 14400153A | -27.510026 | 153.469168 | 71.5 | 8.91 | 31.0 | 22.0 | 4.2 | 62 | 1 | 217.1 | 217.1 | -4.3 | -21.8 |
| Test Hole C | -27.505453 | 153.413512 | 43.5 | 13.60 | 21.4 | 23.0 | 5.4 | 104 | 0.94 | 79.8 | 114 | -2.4 | -11.7 |
| Eight. Mile Swamp | -27.521715 | 153.497049 | nd | na | 0.0 | 24.2 | 7.5 | 115 | 1.47 | 3.2 | 199 | -3.2 | -13.7 |
| Rain | -27.496514 | 153.400427 | na | na | na | na | na | 49 | nd | nd | nd | 0.2 | -16.0 |
| Wonky H. South | -27.451400 | 153.428417 | 0.0 | na | nd | 23.5 | 6.8 | nd | 2.46 | 30.7 | 98 | -4.6 | -23.0 |
| Wonky H. North | -27.450828 | 153.428233 | 0.0 | na | nd | 23.9 | 7.0 | nd | 2.6 | 34.9 | 121 | -4.7 | -23.0 |





**Table A1.** Chemistry data, Lat=Latitude, Lng=Longitude, Water Level= water level below natural surface, Hydr. Head=hydraulic head, Major ion data, molar Cl/Br and Na/Cl ratio. CBE=Charge balance error. (continued)

| RN | $\delta^{13}C$ ‰ | $^{14}C$ pmC | $^{14}C$ error 1 s | $^{3}H$ TU | $^{3}H$ error 1 s | $HCO_3$ mg L$^{-1}$ | F mg L$^{-1}$ | Cl mg L$^{-1}$ | Br mg L$^{-1}$ | $NO_3$ mg L$^{-1}$ | $SO_4$ mg L$^{-1}$ | Na mg L$^{-1}$ | Mg mg L$^{-1}$ |
|---|---|---|---|---|---|---|---|---|---|---|---|---|---|
| Units | | | | | | | | | | | | | |
| 14400051 | -21.2 | 97.65 | 0.27 | 0.56 | 0.04 | 1.7 | bd | 26.91 | 0.09 | 0.51 | 3.74 | 16.51 | 1.75 |
| 14400067 | -21.9 | 98.75 | 0.26 | 0.60 | 0.04 | 0.4 | bd | 19.84 | 0.06 | 0.41 | 3.46 | 12.34 | 1.42 |
| 14400075 | -21.7 | 84.29 | 0.27 | 0.06 | 0.03 | 25.9 | 0.02 | 24.51 | 0.07 | 1.71 | 3.89 | 14.93 | 1.41 |
| 14400087 | -23.9 | 111.38 | 0.33 | 0.58 | 0.04 | 30.8 | bd | 16.51 | 0.06 | 0.16 | 2.54 | 10.90 | 0.88 |
| 14400088 | -25.0 | 103.68 | 0.29 | 1.00 | 0.05 | 0.4 | bd | 15.87 | 0.05 | 0.25 | 0.42 | 9.64 | 1.27 |
| 14400091 | -24.2 | 101.45 | 0.27 | 0.90 | 0.05 | 16 | bd | 15.76 | 0.09 | 0.09 | 1.23 | 9.48 | 1.12 |
| 14400092 | -19.3 | 80.81 | 0.24 | 0.17 | 0.03 | 12 | 0.01 | 23.26 | 0.07 | 0.30 | 2.75 | 13.92 | 0.93 |
| 14400093 | -22.0 | 91.50 | 0.25 | 0.58 | 0.04 | 21.2 | bd | 17.01 | 0.06 | 0.03 | 1.08 | 12.45 | 2.04 |
| 14400094 | -24.0 | 103.85 | 0.27 | 0.93 | 0.05 | 22.4 | bd | 15.89 | 0.07 | 0.04 | 0.68 | 9.85 | 1.37 |
| 14400105 | -24.1 | 84.09 | 0.27 | 0.99 | 0.05 | 44.3 | bd | 30.35 | 0.15 | 0.04 | 21.76 | 21.54 | 3.43 |
| 14400112 | -10.6 | 104.83 | 0.32 | 0.67 | 0.04 | 47.7 | 0.03 | 46.33 | 0.21 | 0.07 | 0.15 | 29.76 | 5.82 |
| 14400114 | -22.2 | 90.99 | 0.28 | 0.34 | 0.03 | 0.4 | bd | 26.95 | 0.09 | 0.39 | 3.44 | 16.07 | 1.74 |
| 14400133 | -21.9 | 95.67 | 0.26 | 0.44 | 0.04 | 0.8 | bd | 16.98 | 0.06 | 0.45 | 2.11 | 9.96 | 1.20 |
| 14400134 | -22.0 | 76.56 | 0.24 | 0.01 | 0.02 | 13 | 0.01 | 18.20 | 0.07 | 0.09 | 3.28 | 11.17 | 1.25 |
| 14400151 | -21.2 | 91.32 | 0.32 | 0.23 | 0.03 | 0.9 | bd | 20.92 | 0.04 | 0.69 | 3.04 | 12.73 | 1.44 |
| 14400152 | -10.7 | 91.87 | 0.31 | 0.49 | 0.04 | 38.8 | 0.02 | 33.72 | 0.14 | 0.29 | 4.42 | 20.84 | 3.09 |
| 14400153A | -21.3 | 92.71 | 0.32 | 0.08 | 0.03 | 0.9 | BD | 22.19 | 0.07 | 0.89 | 3.61 | 13.01 | 1.71 |
| Test Hole C | -17.7 | 62.54 | 0.22 | 0.01 | 0.02 | 20.6 | 0.08 | 19.48 | 0.07 | 0.01 | 1.69 | 15.62 | 2.38 |
| Eight. Mile Swamp | nd | nd | nd | nd | nd | 11.5 | 0.01 | 41.94 | 0.13 | 0.13 | 3.25 | 23.77 | 3.32 |
| Rain | nd | nd | nd | nd | nd | 0 | 0.01 | 19.36 | 0.03 | 3.15 | 5.15 | 11.05 | 1.38 |
| Wonky H. South | -12.0 | 58.83 | 0.21 | 0.12 | 0.03 | 0 | 0.06 | 18.07 | 0.13 | 0.83 | 1.99 | 11.29 | 1.46 |
| Wonky H. North | -12.0 | 80.87 | 0.23 | 0.15 | 0.03 | 0 | 0.06 | 17.83 | 0.17 | 1.09 | 2.04 | 11.23 | 1.36 |



**Table A1.** Chemistry data, Lat=Latitude, Lng=Longitude, Water Level= water level below natural surface, Hydr. Head=hydraulic head, Major ion data, molar Cl/Br and Na/Cl ratio. CBE=Charge balance error. (continued)

| RN | K | Ca | Sr | Ba | Al | Si | Mn | Fe | Cl/Br | Na/Cl | CBE |
|---|---|---|---|---|---|---|---|---|---|---|---|
| Units | mg L⁻¹ | mg L⁻¹ | mg L⁻¹ | mg L⁻¹ | mg L⁻¹ | mg L⁻¹ | mg L⁻¹ | mg L⁻¹ | molar | molar | % |
| 14400051 | 0.57 | 0.28 | 0.01 | 0.00 | 0.01 | 4.52 | 0.00 | 0.01 | 653 | 0.95 | 1.39 |
| 14400067 | 0.47 | 0.24 | 0.01 | 0.01 | 0.01 | 4.54 | 0.02 | 0.02 | 759 | 0.96 | -1.03 |
| 14400075 | 0.98 | 1.72 | 0.01 | 0.02 | 0.10 | 4.79 | 0.05 | 0.22 | 742 | 0.94 | 2.64 |
| 14400087 | 0.25 | 0.25 | 0.00 | 0.00 | 0.02 | 4.25 | 0.01 | 0.19 | 645 | 1.02 | -1.83 |
| 14400088 | 0.39 | 0.42 | 0.01 | 0.00 | 0.46 | 1.56 | 0.00 | 0.27 | 788 | 0.94 | -3.52 |
| 14400091 | 0.35 | 0.39 | 0.00 | 0.00 | 0.19 | 1.99 | 0.00 | 0.17 | 379 | 0.93 | 2.40 |
| 14400092 | 0.63 | 3.00 | 0.02 | 0.03 | 0.01 | 5.51 | 0.10 | 4.80 | 738 | 0.92 | 9.33 |
| 14400093 | 0.36 | 0.76 | 0.01 | 0.01 | 0.09 | 5.05 | 0.04 | 3.31 | 616 | 1.13 | 12.59 |
| 14400094 | 0.35 | 0.27 | 0.01 | 0.00 | 0.80 | 2.36 | 0.00 | 1.56 | 518 | 0.96 | -3.90 |
| 14400105 | 0.40 | 0.56 | 0.02 | 0.01 | 0.88 | 3.15 | 0.01 | 1.73 | 451 | 1.09 | -0.73 |
| 14400112 | 1.10 | 5.69 | 0.05 | 0.00 | 0.07 | 13.20 | 0.18 | 8.40 | 492 | 0.99 | 10.49 |
| 14400114 | 0.54 | 0.25 | 0.01 | 0.01 | 0.01 | 4.62 | 0.00 | 0.01 | 676 | 0.92 | 0.70 |
| 14400133 | 0.37 | 0.07 | 0.00 | 0.01 | 0.05 | 4.62 | 0.00 | 0.01 | 606 | 0.9 | 3.27 |
| 14400134 | 0.70 | 0.37 | 0.01 | 0.02 | 0.02 | 5.40 | 0.03 | 0.01 | 608 | 0.95 | -1.22 |
| 14400151 | 0.74 | 0.27 | 0.01 | 0.01 | 0.01 | 4.72 | 0.01 | 0.00 | 1069 | 0.94 | 1.04 |
| 14400152 | 0.72 | 14.10 | 0.08 | 0.00 | 0.05 | 5.65 | 0.02 | 0.50 | 553 | 0.95 | 7.59 |
| 14400153A | 0.64 | 0.45 | 0.01 | 0.01 | 0.02 | 4.69 | 0.03 | 0.72 | 752 | 0.9 | 2.20 |
| Test Hole C | 1.26 | 3.64 | 0.04 | 0.08 | 0.02 | 5.68 | 0.11 | 0.02 | 634 | 1.24 | 0.29 |
| Eight. Mile Swamp | 0.85 | 2.27 | 0.03 | 0.01 | 0.00 | 3.60 | 0.03 | 0.04 | 725 | 0.87 | 6.10 |
| Rain | 0.83 | 8.18 | 0.01 | 0.01 | 0.00 | 0.06 | 0.00 | 0.00 | 1431 | 0.88 | -7.57 |
| Wonky H. South | 0.52 | 2.12 | nd | nd | nd | 4.43 | 0.01 | 0.00 | 318 | 0.95 | -5.58 |
| Wonky H. North | 0.49 | 1.64 | nd | nd | nd | 4.29 | 0.01 | 0.01 | 237 | 0.96 | -4.14 |





**Table A2.** $^3$H residence times calculated with input function with modern tritium rainfall activities of 1.6 TU and 2.0 TU as well as the average in between the two results. Furthermore, $^{14}$C ages are reported as conventional ages from the laboratory and further ages are calculated using a statistical q factor of 0.85 (Clark and Fritz, 1997) and a correction factor of q =0.95.

| RN | $^3$H (TU) | $^{14}$C (pMC) | PEM ratio | PEM | DM 0.05 | DM 0.5 | EPM | PEM | DM 0.05 | DM 0.5 | EPM |
| --- | --- | --- | --- | --- | --- | --- | --- | --- | --- | --- | --- |
| | | | | | Modern Rainfall input 2 TU | | | | Modern Rainfall input 1.6 TU | | |
| 14400051 | 0.56 | 97.65 | 0.84 | 57 | 58 | 83 | 72 | 57 | 58 | 82 | 72 |
| 14400067 | 0.60 | 98.75 | 0.44 | 58 | 57 | 77 | 69 | 58 | 57 | 76 | 69 |
| 14400075 | 0.06 | 84.29 | 0.20 | 136 | 88 | 290 | 88 | 136 | 88 | 290 | 88 |
| 14400087 | 0.58 | 111.38 | 0.13 | 57 | 57 | 80 | 71 | 57 | 57 | 79 | 71 |
| 14400088 | 1.00 | 103.68 | 0.35 | 37 | 45 | 32 | 34 | 37 | 45 | 27 | 34 |
| 14400091 | 0.90 | 101.45 | 1.24 | 50 | 48 | 41 | 45 | 50 | 48 | 38 | 45 |
| 14400092 | 0.17 | 80.81 | 0.12 | 143 | 75 | 190 | 86 | 143 | 75 | 190 | 86 |
| 14400093 | 0.58 | 91.50 | 0.25 | 57 | 57 | 80 | 71 | 57 | 57 | 79 | 71 |
| 14400094 | 0.93 | 103.85 | 0.37 | 40 | 47 | 38 | 42 | 40 | 47 | 34 | 42 |
| 14400105 | 0.99 | 84.09 | 0.16 | 33 | 45 | 33 | 36 | 32 | 45 | 28 | 36 |
| 14400112 | 0.67 | 104.83 | 0.09 | 50 | 54 | 67 | 64 | 50 | 54 | 66 | 64 |
| 14400114 | 0.34 | 90.99 | 0.24 | 83 | 66 | 130 | 82 | 83 | 66 | 130 | 82 |
| 14400133 | 0.44 | 95.67 | 0.11 | 73 | 62 | 107 | 78 | 73 | 62 | 106 | 78 |
| 14400134 | 0.01 | 76.56 | 0.10 | b.d. | b.d. | b.d. | b.d. | b.d. | b.d. | b.d. | b.d. |
| 14400151 | 0.23 | 91.32 | 0.09 | 124 | 71 | 165 | 85 | 124 | 71 | 165 | 85 |
| 14400152 | 0.49 | 91.87 | 0.45 | 65 | 60 | 95 | 76 | 65 | 60 | 95 | 76 |
| 14400153A | 0.08 | 92.71 | NA | 200 | 84 | 260 | 88 | 200 | 84 | 260 | 88 |
| Test Hole C | 0.01 | 62.54 | NA | b.d. | b.d. | b.d. | b.d. | b.d. | b.d. | b.d. | b.d. |
| Wonky South | 0.12 | 58.83 | NA | 180 | 79 | 220 | 87 | 180 | 79 | 220 | 87 |
| Wonky North | 0.15 | 80.87 | NA | 157 | 77 | 200 | 87 | 157 | 77 | 200 | 87 |





**Table A2.** ³H residence times calculated with input function with modern tritium rainfall activities of 1.6 TU and 2.0 TU as well as the average in between the two results. Furthermore, ¹⁴C ages are reported as conventional ages from the laboratory and further ages are calculated using a statistical q factor of 0.85 (Clark and Fritz, 1997) and a correction factor of q =0.95. (continued)

| RN | ¹⁴C (pMC) | Conv. Ages | PEM | DM 0.5 | DM 0.05 | EPM | PEM | DM 0.5 | DM 0.05 | EPM | PEM | DM 0.5 | DM 0.05 | EPM |
|---|---|---|---|---|---|---|---|---|---|---|---|---|---|---|
| | | | | q=1 | | | | q=0.95 | | | | q=0.85 | | |
| 14400051 | 97.65 | 197 | 100 | 320 | 140 | 2 | 9 | 190 | 9 | 9 | 24 | 49 | 23 | 28 |
| 14400067 | 98.75 | 104 | 3 | 280 | 3 | 3 | 10 | 170 | 10 | 130 | 26 | 49 | 25 | 32 |
| 14400075 | 84.29 | 1412 | 1300 | 1500 | 1300 | 1400 | 860 | 1000 | 910 | 970 | 4 | 270 | 4 | 4 |
| 14400087 | 111.38 | 0 | 20 | 28 | 20 | 20 | 28 | 49 | 26 | 44 | 37 | 49 | 42 | 44 |
| 14400088 | 103.68 | 0 | 10 | 10 | 10 | 10 | 17 | 20 | 17 | 17 | 32 | 49 | 30 | 44 |
| 14400091 | 101.45 | 0 | 7 | 7 | 7 | 7 | 14 | 15 | 14 | 14 | 29 | 49 | 28 | 44 |
| 14400092 | 80.81 | 1761 | 1600 | 1900 | 1700 | 1800 | 1300 | 1400 | 1200 | 1300 | 420 | 450 | 460 | 430 |
| 14400093 | 91.50 | 734 | 640 | 750 | 710 | 730 | 360 | 380 | 360 | 310 | 15 | 17 | 15 | 15 |
| 14400094 | 103.85 | 0 | 10 | 10 | 10 | 10 | 17 | 21 | 17 | 17 | 32 | 49 | 31 | 44 |
| 14400105 | 84.09 | 1432 | 1300 | 1500 | 1300 | 1400 | 960 | 1000 | 930 | 990 | 60 | 280 | 4 | 4 |
| 14400112 | 104.83 | 0 | 12 | 160 | 80 | 11 | 18 | 24 | 18 | 19 | 33 | 49 | 32 | 44 |
| 14400114 | 90.99 | 780 | 730 | 800 | 740 | 780 | 370 | 410 | 400 | 360 | 14 | 130 | 14 | 14 |
| 14400133 | 95.67 | 366 | 480 | 410 | 410 | 370 | 6 | 230 | 6 | 6 | 22 | 35 | 21 | 22 |
| 14400134 | 76.56 | 2207 | 2100 | 2500 | 2200 | 2300 | 1600 | 2000 | 1700 | 1800 | 750 | 890 | 810 | 860 |
| 14400151 | 91.32 | 750 | 720 | 770 | 720 | 750 | 360 | 390 | 370 | 330 | 15 | 16 | 15 | 15 |
| 14400152 | 91.87 | 701 | 700 | 710 | 680 | 700 | 220 | 360 | 320 | 260 | 16 | 120 | 16 | 110 |
| 14400153A | 92.71 | 626 | 610 | 640 | 620 | 630 | 100 | 320 | 140 | 170 | 17 | 20 | 17 | 17 |
| Test Hole C | 62.54 | 3879 | 4100 | 5000 | 4200 | 4500 | 3600 | 4300 | 3700 | 3900 | 2500 | 3000 | 2600 | 2700 |
| Wonky South | 58.83 | 4384 | 4800 | 5900 | 4900 | 5200 | 4300 | 5200 | 4300 | 4600 | 3100 | 3700 | 3200 | 3300 |
| Wonky North | 80.87 | 1755 | 1600 | 1900 | 1700 | 1800 | 1200 | 1400 | 1200 | 1300 | 390 | 450 | 450 | 420 |