# Peer review of "Groundwater mean residence time of a sub-tropical barrier sand island"

_Hydrology and Earth System Sciences, 2019_

## Referee Comment (RC1) · Anonymous Referee #1 · 12 Oct 2019

General comments

The paper nicely shows how environmental tracers provide information on groundwater ages and on the functioning of groundwater systems. The issue addressed in this paper – hydrogeology of sand barrier islands – is of global relevance. Presented results contribute to the understanding of this important groundwater resource and its sustainability under conditions of population growth and climate change.

The paper is well written and structured. The background, environmental context and purpose of this work are clearly explained, followed by a well-balanced and comprehensive presentation of the tracer and modeling approaches. The data and modeling results are presented in a clear manner.

[Figure]

A question arises, however, what is the main contribution of this work? The discussion and conclusions sections highlight two results of the tracer exercise: (i) spatial distribution of groundwater ages identifies groundwater flow paths, (ii) low permeability units are responsible for long MRTs.

Ad (i). It is not surprising that the distribution of MRTs conforms the presumed picture of a "groundwater mound" with uniform recharge and dominant unconfined flows perpendicular to island's axis simply because the LPM was selected accordingly. What is an added value coming from the application of tracers? Furthermore, in my opinion the dependence of tritium concentrations on the distance from island's centre and bore depth as shown in Fig. 5 is not obvious. There is a lot of scatter in these data. Could this scatter and other discrepancies in MRT results be due to a more complex than assumed structure of groundwater flow? The authors mention "minor" longitudinal flows due to the complex topography not discussing this question further. This is understandable as the study is based on one transect only, which might be its main shortcoming. If the topography is complex, then could the longitudinal flows occur? For example, do the dunes have any dominant orientation (due to prevailing wind conditions in the past), or is the island's relief completely irregular?

Ad (ii). The only direct evidence for that comes from a comparison of MRTs in two bores. How relevant is this finding to the overall flow patterns? Again, how widespread are peat deposits along the island? Are they continuous or patchy?

Unfortunately the conclusions section lacks the clarity and accuracy of earlier sections. The contribution and added value of the paper should be more clearly expressed.

Specific comments

Given some ambiguity in the understanding of terms "residence time" and "transit time" it is advisable not to use them interchangeably.

Lines 83-90. A more detailed description of topography (with a picture showing the

landscape?) will help to comprehend the natural setting – see the general comments. Is the vegetation cover of dunes continuous?

Lines 92-93. How do precipitation and evapotranspiration vary seasonally?

Figure 4. Some of groundwater samples seem to be affected by evaporation. Could the low d-excess waters indicate recharge from wetlands? On the other hand, two samples have very high d-excess values. Do the stable isotopes have in this case any potential as indicators of recharge areas and flow paths?

Line 261. The decrease of tritium activities with distance from centre is obvious for the first 2500 m only – see general comments.

Lines 230-1. What is a possible source of carbonate in this presumably carbonate-free geological setting? Are there any secondary carbonate deposits associated with paleosoils? If not, does the bedrock contain any carbonate? In the latter case, higher carbonate dissolution is related to groundwater contact with the bedrock or groundwater discharge from the basement.

Line 325. What is the actual carbon isotopic signature of soil DIC and how was it derived?

Technical comments

Figure 5.A is not mentioned in text.

Line 267. Test hole or testhole?

---

## Referee Comment (RC2) · Anonymous Referee #2 · 25 Oct 2019

General comments The authors describe the mean residence times of groundwater in a fresh-water lens on a barrier island in Australia. The paper is generally well written, easy to follow and the methods and their application is straightforward. In general, it is a contribution worthwhile publishing. There are, however, a few points that need to be addressed:

The authors tend to focus on rather recent literature to introduce concepts (e.g. lines 38-39, 40-41 but also elsewhere). This undervalues the contributions of the people who developed these concepts in the first place. Priority should be given to the older literature.

What is surprisingly almost completely missing is a comparison of the obtained data to other barrier islands, of which there are many worldwide. Several studies have studied

the MRT (or age patterns) on barrier islands/dune areas in the Netherlands (Stuyfzand 1993) and on the German barrier islands Borkum, Spiekeroog, Langeoog and Baltrum (search for authors Holt, Seibert, Greskowiak, Massmann, Wiederhold, Post, Houben, Stoeckl etc.).

A recommendation would be to try to use the analytical model by Fetter (1972, the one with the impermeable basement) to try to recreate the lens shape depicted in Figure 9 with the parameters the authors propose. The age patterns could also be checked against the analytical models by Vacher (1988) and Chesnaux & Allen (2013).

Screen lengths of the observation wells are not given but may be an important factor. Considering the low tritium concentrations found, results can be easily affected by mixing, if samples are taken from long-screened wells. Please add info!

Specific comments Line 1-2: groundwater use and over-abstraction are related L25: do not agree, the barrier islands along the North Sea shore have no perched aquifers, this might be true for Australia but not necessarily for all barrier islands L46: I would disagree, there was steady stream of publications on the German barrier islands in the last few years, especially on Borkum, Spiekeroog, Langeoog and Baltrum. Hardly any of the publications are cited in this manuscript (except for Röper et al.), therefore, the statement on the poor understanding of such systems is not valid. Several of these studies explicitly address the topic of residence times (and also of groundwater climate archive). L130: what was the screen length? L162ff: I wonder why tritium-helium was not considered, as it frees you from many of the model assumptions of lumped parameter models such as the PFM et al.? L241: maybe better to use dissolved oxygen concentrations instead of ORP L245, 246: two decimal places really needed/valid? L365: please avoid colloquial terms like "coffee rock"

---

## Author Response (AR1)

**Reply to editor for manuscript hess-2019-304 Groundwater mean residence time of a sub-tropical barrier sand island, by Hofmann et al.**

Dear Dr Stumpp,

We thank you for the possibility to resubmit our revised version of the manuscript "Groundwater mean residence time of a subtropical barrier sand island".

The reviewer's comments were very helpful and we have incorporated all your recommendations; (i) highlight the new findings of the study compared to others, (ii) how to conclude from results at the observation points to the entire flow system, (iii) comparison of results with findings from other barrier islands.

The novelty of this study lies in the application of environmental tracers to barrier island that are distinctly different from the barrier island that were investigated in the past and that reviewer 2 pointed out in the way that the past investigation were conducted on very young (mostly Holocene) sand island that had not developed distinct stratigraphic unit. The sand masses along the east coast of Australia are by comparison much older, reaching back to the early Pleistocene. Their internal structure has developed by vegetation, pedogenesis and dune movement. The result is a much more complex dune stratigraphy with a variety of hydrological units which leads to the inconsistent mean residence time (MRT) distribution and in general quite long MRT compared to the studies mentioned above (suggestion I + III).

We highlighted this difference and compared our results to other environmental tracer studies on barrier islands in the revised manuscript and have also added the studies that were relevant according to reviewer 2 comments.

As for the second suggestion (suggestion II), transferring the finding from a transect to the entire flow system is a qualitative observation. There are studies (e.g. Ellerton et al., 2018) that demonstrate that the complex dune stratigraphies are common landforms all along the east coast of Australia, and in other part of the world (e.g. Netherland; Stuyfzand, 2013) with older dune systems. A quantitative estimation of the entire flow system was never our aim as much more detailed spatial information on the dune stratigraphy is necessary. Our aim is to highlight that these systems have been simplified in the past and MRT were hugely underestimated.

We have reviewed all the technical comments from the reviewer and have changed the text accordingly. The application of an analytical model was not the aim for this study either and we think we have sufficiently justified the reasons why in the response to the reviewers.

All changes are highlighted in red in the revised version..

We hope that we satisfy the requirement to have this manuscript finalized for publication.

Kind regards,

Harald Hofmann et al.

**Response to review's comments**

**Reviewer 1:**

**General comments**
Reviewer 1: The paper nicely shows how environmental tracers provide information on groundwater ages and on the functioning of groundwater systems. The issue addressed in this paper – hydrogeology of sand barrier islands – is of global relevance. Presented results contribute to the understanding of this important groundwater resource and its sustainability under conditions of population growth and climate change. The paper is well written and structured. The background, environmental context and purpose of this work are clearly explained, followed by a well-balanced and comprehensive presentation of the tracer and modeling approaches. The data and modelling results are presented in a clear manner.
Response: We thank the reviewer for the overall positive comments on this manuscript and pointing out the global relevance of this study.

Reviewer 1: A question arises, however, what is the main contribution of this work? The discussion and conclusions sections highlight two results of the tracer exercise: (i) spatial distribution of groundwater ages identifies groundwater flow paths, (ii) low permeability units are responsible for long MRTs.

Response: As we discuss in section 5.2 and following, the paper demonstrates that groundwater flow in coastal sand masses is much more complex that previously thought. Using the tracer data combined with field observations to construct conceptual models that is outlined in the discussion is a major contribution to the overall understanding of groundwater flow in coastal sand masses. This conceptualisation is important for modelling exercises and for determining the sustainable use or groundwater and/or the potential impacts of groundwater use. We will emphasise these points more when we do the revision.

Studies investigating the spatial distribution and total impact of lower permeability units in coastal sand dunes would certainly help to better understand the hydrogeology.

Reviewer 1: Ad (i). It is not surprising that the distribution of MRTs conforms the presumed picture of a "groundwater mound" with uniform recharge and dominant unconfined flows perpendicular to island's axis simply because the LPM was selected accordingly. What is an added value coming from the application of tracers? Furthermore, in my opinion the dependence of tritium concentrations on the distance from island's centre and bore depth as shown in Fig. 5 is not obvious. There is a lot of scatter in these data. Could this scatter and other discrepancies in MRT results be due to a more complex than assumed structure of groundwater flow?

Response: Figure 5 is key to understanding that flow patterns are relatively complex. There is a general trend from the centre of the island (groundwater mount to both coastlines, which is what one would expect for sand islands in general. However, as discussed in the paper (section 5.1) the tritium distribution and calculated MRTs are less regular. This indicates that the flow paths are more complex than assumed in some of the previous groundwater models (Leach and Gallaghar, 2008). The tritium data thus adds significant value in providing a more valid conceptualisation for any future modelling exercises than could be determined from the groundwater heads and hydraulic properties alone. This is discussed in the paper (section 5.4 & 5.5) and we will emphasise this in the revision.

Reviewer 1: The authors mention "minor" longitudinal flows due to the complex topography not discussing this question further. This is understand-able as the study is based on one transect only, which might be its main short coming. If the topography is complex, then could the longitudinal flows occur? For example, do the dunes have any dominant orientation (due to prevailing wind conditions in the past), or is the island's relief completely irregular?

Response: According to the literature and field observations, the dunes have formed from predominantly onshore south-easterly winds. The aquifer and dune systems are elongated north-south and, while there is the possibility of some longitudinal flow, the available groundwater head data indicate that flow is predominantly eastward and westward from the centre of the island. The transect of state government groundwater bores were installed parallel to the dominant groundwater flow direction. While we recognised the possibility of

longitudinal flow, there are very few bores to the north of the transect and access is limited to the south due to mining lease access restrictions. A larger more spatially distributed sampling network would have been preferable; however, we are confident that we have sampled perpendicular to the main direction of groundwater flow and thus that the pattern of MRTs is representative.

Reviewer 1: Ad (ii). The only direct evidence for that comes from a comparison of MRTs in two bores. How relevant is this finding to the overall flow patterns? Again, how widespread are peat deposits along the island? Are they continuous or patchy? Unfortunately, the conclusions section lacks the clarity and accuracy of earlier sections. The contribution and added value of the paper should be more clearly expressed.

Response: There was only the one nested bore site available, which we agree limits the certainty of this conclusion. As peat layers are very common across the island and occur around most wetland and lake systems, this is potentially an important point. We will revise the conclusions to clarify that this is an isolated observation but one that is important in understanding the overall hydrogeology.

**Specific comments**

Reviewer 1: Given some ambiguity in the understanding of terms "residence time" and "transit time" it is advisable not to use them interchangeably.

Response: We thank the reviewer for the comment and will adopt "residence time" throughout the text.

Reviewer 1: Lines 83-90. A more detailed description of topography (with a picture showing the landscape?) will help to comprehend the natural setting – see the general comments. Is the vegetation cover of dunes continuous?

Response: The vegetation cover of the dunes is continuous apart from wetlands, lakes, tracks and mining operations. We will note that in the revision. We can add a picture of the environment but would prefer to add it to the supplementary materials.

Reviewer 1: Lines 92-93. How do precipitation and evapotranspiration vary seasonally?

Response: There are slight differences in precipitation and evapotranspiration between winter and summer. In general, south-east Queensland has dry winters and most of the rainfall occurs during the summer months. The mainland variability is dampened on the coast with possible rainfall across the year. Evapotranspiration is lowest in winter (June/July) and highest in summer (December/January). We will add this information to the revised text.

Reviewer 1: Figure 4. Some of groundwater samples seem to be affected by evaporation. Could the low d-excess waters indicate recharge from wetlands? On the other hand, two samples have very high d-excess values. Do the stable isotopes have in this case any potential as indicators of recharge areas and flow paths?

Response: We agree that some of the groundwater shows signs of evaporation. It is also true that evaporation does occur in the lakes at the wetlands. However, as discussed in the Discussion, the wetlands and lakes probably hold back water from directly percolating to the regional aquifer. Most lakes and wetlands are underlain by perched water tables that are not directly connected to the main aquifer. With regards to the two samples with the high d-excess, we were not able to find a systematic connection between these samples and recharge patterns.

Reviewer 1: Line 261. The decrease of tritium activities with distance from centre is obvious for the first 2500 m only – see general comments.

Response: As discussed above, this indicates the complexity of the dune systems and that flow patterns are not as simple as previously thought, which is one of the main conclusions of the paper. We will add a few more sentences to explain the data better and emphasise this finding in the Discussion.

Reviewer 1: Lines 230-1. What is a possible source of carbonate in this presumably carbonate-free geological setting? Are there any secondary carbonate deposits associated with paleosoils? If not, does the bedrock contain any carbonate? In the latter case, higher carbonate dissolution is related to groundwater contact with the bedrock or groundwater discharge from the basement.

Response: As mentioned in the text (lines 327-328), carbonate dissolution is minor and has probably affected only a few samples. The general hydrogeochemical environment has mostly lower pH values and secondary carbonates in topsoils are very unlikely. We agree that paleosoils are likely sources of DIC. An explanation for higher d13C that we haven't considered yet is sea spray, especially for those bores in the east close to the ocean. We will add this to the discussion. The correction for DIC dissolution produces slightly younger MRTs, however, the MRTs are still longer than were initially expected probably due to the complex flow patterns.

Reviewer 1: Line 325. What is the actual carbon isotopic signature of soil DIC and how was it derived?

Response: The soil DIC 13C values are based on literature values as mentioned earlier in the paper (lines 272). We will amend this here and be more specific that the values are estimated.

**Technical comments**

Reviewer 1: Figure 5.A is not mentioned in text.

Response: We will add a reference to figure 5A in line 261.

Reviewer 1: Line 267. Test hole or testhole?

Response: We will correct this to "testhole"

**Reviewer 2:**

**General comments:**

Reviewer 2: The authors describe the mean residence times of groundwater ina fresh-water lens on a barrier island in Australia. The paper is generally well written, easy to follow and the methods and their application is straightforward. In general, it is a contribution worthwhile publishing. There are, however, a few points that need to be addressed: The authors tend to focus on rather recent literature to introduce concepts (e.g. lines 38-39, 40-41 but also elsewhere). This undervalues the contributions of the people who developed these concepts in the first place. Priority should be given to the older literature.

What is surprisingly almost completely missing is a comparison of the obtained data to other barrier islands, of which there are many worldwide. Several studies have studied the MRT (or age patterns) on barrier islands/dune areas in the Netherlands (Stuyfzand 1993) and on the German barrier islands Borkum, Spiekeroog, Langeoog and Baltrum (search for authors Holt, Seibert, Greskowiak, Massmann, Wiederhold, Post, Houben,Stoeckl etc.).

Response: We thank the reviewer 2 for the positive feedback and welcome suggestions. Our intent was not to give a full review on the salt water/freshwater interface but only to point out that there is a vast amount of literature.

We will add extra citations by Holt et al., Seibert et al., Holding and Allen, and Houben et al. on the topic, which have sections on residence times on barrier islands. Many of the other publications mentioned by reviewer 2 is work on the salt water / freshwater interface and biological as well as bio-geochemical processes.

Reviewer 2: A recommendation would be to try to use the analytical model by Fetter (1972, the one with the impermeable basement) to try to recreate the lens shape depicted in Figure 9 with the parameters the authors propose. The age patterns could also be checked against the analytical models by Vacher (1988) and Chesnaux & Allen (2013). Screen lengths of the observation wells are not given but may be an important factor. Considering the low tritium concentrations found, results can be easily affected by mixing, if samples are taken from long-screened wells. Please add info!

Response: These models and others (such as those by Post) have uniform geometries and hydraulic properties. As noted in Fetter 1972, modelling of the freshwater lens in a real island requires that the detailed geometry of the island be taken into account and recharge rates to be well known. Even with these parameters, the assumptions of uniform hydraulic conductivity and steady state conditions will cause some differences with the field examples. This paper is not focussed on modelling and constructing such an analytical model is a contribution it itself (as is evident from the many papers that have done this).

As noted below, these bores are groundwater observation bores not abstraction bores and have screens of 1.5 m are at the bottom above a 1.5 m sump. Given these small screens, mixing due to the bore sampling multiple groundwater sources is probably not major.

**Specific comments**

Reviewer 2: Line 1-2: groundwater use and over-abstraction are related

Response: We will change this to over-abstraction only in the revised version

Reviewer 2: L25: do not agree, the barrier islands along the North Sea shore have no perched aquifers, this might be true for Australia but not necessarily for all barrier islands

Response: We thank the reviewer for this comment and will specify that perched aquifer systems are often observed in subtropical coastal sand islands.

Reviewer 2: L46: I would disagree, there was steady stream of publications on the German barrier islands in the last few years, especially on Borkum, Spiekeroog, Langeoog and Baltrum. Hardly any of the publications are cited in this manuscript (except for Röper et al.), therefore, the statement on the poor understanding of such systems is not valid. Several of these studies explicitly address the topic of residence times (and also of groundwater climate archive).

Response: We have looked at the suggested publications and will add the references to provide more global context. Some of those studies have residence time estimations, however, all of them are in a completely different climatic and geological setting. We will be clearer in the revised version that this is one of the first studies on subtropical barrier sand islands.

Reviewer 2: L130: what was the screen length?

Response: The screen lengths were 1.5 m. These bores are groundwater observation bores not abstraction bores and screen sections are at the bottom above a 1.5m sump. We will add this information to a revised version of the manuscript.

Reviewer 2: L162ff: I wonder why tritium-helium was not considered, as it frees you from many of the model assumptions of lumped parameter models such as the PFM et al.?

Response: Tritium/He was mainly developed to allow the continued use of tritium following the diminishing of the high tritium activities arising from the atmospheric nuclear tests (the tritium bomb-pulse). The development of low-level tritium analysis also allows achieves this. This is important in the southern hemisphere as the bomb pulse was far lower than the northern hemisphere and has long since decayed; hence most water has tritium activities that are significantly lower than rainfall (<3 TU in Australia). It is true that the tritium/He method does not require the tritium input function to be known. However, unless piston flow is assumed, MRTs still have to be calculated via lumped parameter models (or some model that allows dispersion and mixing within the aquifer to be accounted for). The tritium input function in Brisbane (adjacent to Stradbroke Island) is well known (Tadros et al., 2014) and this is not the major uncertainty in calculating MRTs. Another important point is that the 3H/He method is sensitive to He degassing (during sampling or in the aquifer which is unconfined). Analysing the total He and differentiating its sources (excess air, terrigenic) from He derived from tritium decay is not without problems. Being able to analyse low level 3H bypasses all those non trivial complications. More pragmatically, the analyses were conducted at ANSTO via funding specific to that facility and there are low-level tritium but not tritium/He capabilities available.

Reviewer 2: L241: maybe better to use dissolved oxygen concentrations instead of ORP

Response: Both are indicators for reduced and oxidising environment. Our aim is to show the differences in redox potential and think ORP is adequate to use.

Reviewer 2: L245, 246: two decimal places really needed/valid?

Response: The two decimal places are not needed and we can round the number.

Reviewer 2: L365: please avoid colloquial terms like "coffee rock"

Response: "Coffee Rock" is a common term for indurated sands with some organic content. It appears black and the term is used along the coast of South-East Queensland and northern New South Wales. The term was explained in the introduction of the text and we think it is adequate to use it in the conclusions.

---

## Author Response (AR2)

Response to final decision on manuscript hess-2019-304.

Dear Dr. Stumpp,

Many thanks for the positive news. We have adapted you suggestions have moved the new part in the conclusions to the discussion. We also have deleted "by contrast" from the second paragraph in the conclusion section.

We hope that the manuscript is now in a state to be published.

Kind regards,

Harald Hofmann et al.